# RETHINKING THE DEPENDENCE BETWEEN GRADIENTS AND THE INITIAL POINT IN DEEP LEARNING

## ABSTRACT

Despite the considerable advancements in Deep Neural Networks (DNNs), their intrinsic opacity remains a challenge from their foundational design. In this study, we elucidate a novel phenomenon wherein the representation of cumulative gradients (the aggregate changes in iterative gradients) exhibits a certain independence from the initial computation point of the gradients. This implies that learned gradients can be assigned to other arbitrarily initialized yet well-trained neural networks, while retaining a comparable representation to the original network. This suggests that the cumulative gradients can be assigned to other arbitrarily initialized but adequately trained neural networks, maintaining a representation like the original one. This occurrence is counterintuitive and can not be well explained via existing optimization theories. Additionally, we observe that the learned model weights can also be reassigned to different neural networks. In essence, these learned gradients can be viewed as a neural network with analogous representations. Futhermore, this reassignment of gradients and model weights can potentially mitigate catastrophic forgetting when learning multi-tasks. We provide a theoretical framework to support this claim. Our extensive experiments clearly illustrate this phenomenon and its potential to mitigate catastrophic forgetting.

## 1 INTRODUCTION

Deep Neural Networks (DNNs) have demonstrated remarkable success in a variety of machine learning tasks, including but not limited to computer vision Simonyan & Zisserman (2015); He et al. (2016), natural language processing Devlin et al. (2019), and robotics Duan et al. (2017); Plappert et al. (2018). Nonetheless, the learning processes of DNNs, particularly with optimization methods such as Stochastic Gradient Descent (SGD) Wijnhoven & de With (2010) and Adam Kingma & Ba (2015), remain theoretically unexplained. This lack of understanding extends to the organization of the internal structures of DNNs. Typically, these unresolved issues contribute to the characterization of DNNs as "black boxes" Alain & Bengio (2017).

In an attempt to better comprehend these "black boxes," researchers Alain & Bengio (2017) have utilized linear classifiers as "probes" to enhance intuition about DNN layers. Other studies have demonstrated the role and dynamics of each layer Yosinski et al. (2014) and provided innovative visualization techniques to offer insights into the layers of DNNs Zeiler & Fergus (2014). Moreover, previous research Tishby & Zaslavsky (2015) has analyzed the mutual information values between input and output variables for each DNN layer. It was concluded that the primary objective of the learning process at each DNN layer is related to the information bottleneck tradeoff between the compression of input information and prediction. Subsequent research Shwartz-Ziv & Tishby (2017) built on this concept, demonstrating that DNNs spend most of the learning process epochs on compressing input information rather than memorizing labels.

This understanding provides some plausible explanations for the well-known generalization problem in deep learning Zhang et al. (2017), a problem that cannot be addressed by conventional regularization and minimal generalization errors. This issue is attributed to the intriguing phenomenon that DNNs using stochastic gradient methods can easily memorize randomly labeled data. Despite its counterintuitive nature, this phenomenon elucidates the generalization capabilities of DNNs, informing future researches Zhang et al. (2018); Mathis et al. (2018); Gu et al. (2018).

Typically, DNNs employ standard optimization methods, such as SGD and Adam, to iteratively update parameters. For instance, during training, SGD estimates the gradients at the initial point using randomly selected examples Wijnhoven & de With (2010). Consequently, it is intuitive to believe that the gradients heavily rely on the initial point of computation, and that optimal representations can only be obtained through a combination of both. However, our experiments revealed that the primary factor influencing the representations is embedded within the structure of the gradients and can be expressed independently of the initial point.

In this study, we introduce an intriguing phenomenon: the representation of cumulative gradients – the aggregated changes of numerous iterative gradients – does not solely rely on the initial point from which the gradients are computed. Consequently, these cumulative gradients can be assigned to newly initialized, well-trained neural networks (of identical architecture), yielding similar representations to the original network. Furthermore, these cumulative gradients can be treated as a new model, encapsulating representations comparable to the original network. Like Zhang et al. (2017), this phenomenon is counterintuitive and cannot be explained using conventional theories. Under traditional neural network optimization theory Nguyen & Hein (2017); Li et al. (2018), all local minima are approximations of the global optimum, implying that virtually all local minima are global optima. However, this principle seems to contradict the above-described phenomenon. This is because the cumulative gradients can be assigned to different points, which are not necessarily close to each other, while still preserving the representations.

In this work, we shed light on the novel phenomenon of cumulative gradients. This phenomenon challenges the traditional perspective on non-convex optimization of neural networks and provides further insight into the "black boxes" of DNNs. Our main contributions are as follows:

- We introduce a compelling phenomenon: the representation of corresponding gradients does not entirely depend on the initial point from which the gradients are computed. This discovery provides valuable insights into understanding the "black boxes" of DNNs.

- Like cumulative gradient assignment, model fusion (assigning the weights of one network to another) is another method to transfer learned representations to different models, potentially leading to superior representations.

- Both cumulative gradient assignment and model fusion can mitigate the catastrophic forgetting problem in neural networks. We provide a theoretical analysis to support this claim.

- Extensive experiments have clearly illustrated this phenomenon across a multitude of different types of neural networks, including 5-layer convolution networks and ResNets.

## 2 RELATED WORK

The study of neural network representation power has a rich history Cybenko (1989); Delalleau & Bengio (2011), with a multitude of research focusing on applying universal approximation theorems for DNNs Mhaskar (1993); Telgarsky (2016); Cohen & Shashua (2016); Eldan & Shamir (2016). These methods characterize the range of mathematical functions that DNNs can express from a population-level perspective. However, due to the "black-box" nature of neural networks, traditional techniques often struggle to interpret them. For instance, the crucial generalization problem of deep learning cannot be clarified by conventional regularization and small generalization errors Zhang et al. (2017). Such issues primarily arise from the representation problem caused by noisy labels during training. In contrast, this paper emphasizes the representation of cumulative gradients and explores how this representation changes when assigned to different models, which are not the initial points for computing the gradients.

The initialization of neural networks is widely recognized as a crucial factor in training an effective network. An appropriate initialization scheme can facilitate the training of basic CNNs, exceeding ten thousand layers, without the need for additional architectural strategies Xiao et al. (2018). Novel methods He et al. (2015) for nonlinear activation functions, inspired by the same premise, have been proposed. Notably, initializing the weights from the orthogonal group has demonstrated accelerated convergence Hu et al. (2020).

Beyond appropriate initialization values, parameter optimization is a key to effective network training. Stochastic Gradient Descent (SGD) and its optimized versions have been successfully employed to train deep neural networks across a range of machine learning tasks. Simple averaging of multiple points along the SGD trajectory, coupled with a cyclical or constant learning rate, typically results in superior generalization compared to traditional training Izmailov et al. (2018). Moreover, phase-wise parameter averaging has been proposed to enhance SGD Kobayashi (2021). Trainable weight averaging further optimizes the averaging coefficients, significantly reducing the estimation error of stochastic weight averaging Li et al. (2022). Another novel optimization strategy for some SGD variants begins with batch size. Post-local SGD has been proposed to address the generalization issue of large-batch training, enabling scaling of training to a much higher number of parallel devices and reaching flatter minima Lin et al. (2020).

The above methods aim to optimize the training process from various stages and perspectives, with the ultimate goal of producing models with more information and superior generalization capabilities. In this work, we find that the representation of a well-trained neural network comprises two components: one from the initial point and the other from the cumulative gradients. Even though the cumulative gradients are computed from the initial point, their representation does not entirely depend on the initial point.

## 3 THE REPRESENTATION OF GRADIENTS

### 3.1 ASSIGNING THE ACCUMULATIVE GRADIENTS TO DIFFERENT MODELS

This section aims to explore the relationship between gradients and their corresponding initial points, as well as their impact on the representations within a given neural network. We employ a methodology of empirical randomization tests to provide experimental evidence towards this goal. This empirical evidence suggests that the cumulative gradients can be assigned to different neural networks, given that they share the same architecture, enabling these networks to hold the same or most of the representations as the original one.

First, we introduce the notations and primary definitions. During the standard training process of a neural network, the goal of cumulative gradient is to to minimize the information in the weights $I(z; n)$ by either compressing the nuisances variable $n$ or injecting new information from new samples into the weights Shwartz-Ziv & Tishby (2017). According to traditional optimization theory, in the parameter space of a neural network, one or several local optima or global optima exist that can represent a given task $A$ Nguyen & Hein (2017); Li et al. (2018). Specifically, for a set of randomly initialized networks $\{\theta_0, \theta_1, \theta_2, \ldots, \theta_m\}$, they can learn task $A$ via the stochastic gradient optimization method. Formally, the training processes are as follows:

$$\theta'_0 = \theta_0 - \beta \nabla_{\theta_0} \sum_{(x,y) \in \mathcal{D}_A} \mathcal{L}_A, \tag{1}$$

$$\theta'_1 = \theta_1 - \beta_1 \nabla_{\theta_1} \sum_{(x,y) \in \mathcal{D}_A} \mathcal{L}_A, \tag{2}$$

$$\theta'_2 = \theta_2 - \beta_2 \nabla_{\theta_2} \sum_{(x,y) \in \mathcal{D}_A} \mathcal{L}_A, \tag{3}$$

$$\ldots$$

$$\theta'_m = \theta_m - \beta_m \nabla_{\theta_m} \sum_{(x,y) \in \mathcal{D}_A} \mathcal{L}_A, \tag{4}$$

where $\mathcal{D}_A$ is the dataset of task $A$, $(x, y)$ represents the corresponding data point sampled from $\mathcal{D}_A$, $\beta$ is the learning rate, and $\mathcal{L}_A$ is the loss function of task $A$. Note that, at the beginning of the epoch, $\theta_i$ will be updated by a batch of data from $\mathcal{D}_A$ to $\theta'_i$, where $i \in \{0, 1, 2, \ldots, m\}$. Then, the new weights $\theta'_i$ will be updated in the next iteration. For simplicity and clarity, we use the one-step iteration $\theta_i - \nabla_{\theta_0} \sum_{(x,y) \in \mathcal{D}_A} \mathcal{L}_A$ to denote it. Henceforth, we adopt the same formulation. The conclusions in Equation equation 4 have been supported by numerous empirical results Nguyen & Hein (2017); Li et al. (2018).

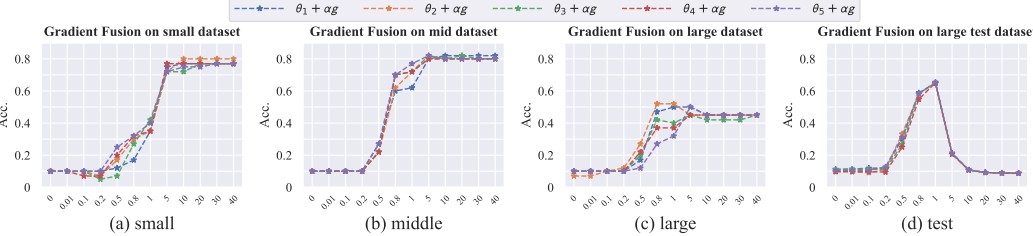

Figure 1: Empirical results for gradient assignment with different randomly initialized models. (a) depicts the results of assigning small-scale cumulative gradients to various randomly initialized models. (b) and (c) illustrate the results on medium and large scales, respectively. (d) presents the results of testing the gradient-assigned model on the test set. The vertical axis represents the accuracy of the gradient-assigned model tested on the CIFAR10 dataset, while the horizontal axis indicates different values of the scaling factor $\alpha$.

Let $g$ represent the cumulative gradients of $\theta_0$:

$$g = -\beta \nabla_{\theta_0} \sum_{(x,y) \in \mathcal{D}_A} \mathcal{L}_A = \theta_A - \theta_0. \tag{5}$$

Interestingly, we found through experimentation that instead of directly computing the gradient with the corresponding position (e.g., $\theta_i$) to update the model, the cumulative gradients (e.g., $g$) can be directly assigned to any other networks $\theta_i$ and still maintain (most of) the learned representation of task $A$:

$$\tilde{\theta}_i = \theta_i + \alpha g, \tag{6}$$

where $\alpha$ is a scale factor. This observation, although counterintuitive, is intriguing and opens up new areas of exploration.

### 3.2 EMPIRICAL RESULTS FOR GRADIENT ASSIGNMENT

We carried out several experiments with varying initial random points to investigate the relationships between the gradients and their respective initial points. Additionally, we conducted tests using different well-trained models, all of which yielded consistent results.

The experiments were conducted using the CIFAR10 dataset Krizhevsky (2009), which is a simple 10-way image classification dataset. To analyze the impact of cumulative gradients more clearly, we conducted experiments at three different scales: small, medium, and large. Specifically, the small-scale experiments were trained on 4 samples per class randomly selected from the dataset. Correspondingly, the medium-scale experiments were trained on 100 randomly selected samples (including the 4 samples from the small-scale experiment) per class, while the large-scale experiments involved training on the entire dataset. For model testing, we use the training samples from the small-scale experiments to test and thus directly reflect the representations implied by the cumulative gradients. Additionally, we present the results of the test set with cumulative gradients computed on the entire training set.

To mitigate the influence of the neural network architecture and clearly showcase the effectiveness of the proposed method, we deliberately opted for a simple, shallow, and widely adopted neural network structure without any architectural modifications throughout all our experiments. Specifically, in Section 3 and Section 4, we employed the original convolution network as our network architecture. In Section 5, we utilized the residual structures He et al. (2016). For further insights into the experimental settings, please refer to Section A in Appendix.

From Figure 1, we find that when assigning the cumulative gradients $g$ to different randomly initialized models $\theta_i$, the new model $\theta_n = \theta_i + \alpha g$ still results in robust representation. The specific values of Figure 1(a) are shown in Table 1. As seen in the table, the classification accuracy of $\theta_n$ on the CIFAR10 dataset increases with the scale factor $\alpha$. When $\alpha$ exceeds 5, the new models achieve an accuracy greater than 0.7. This accuracy is sufficient to demonstrate that $\theta_n$ contains categorical information about the dataset, but it is interesting to note that $\theta_n$ is stitched together from irrelevant

Table 1: Assigning the small-scale accumulative gradients $g$ to different random initialized models, where $\theta_i$ is random initialized model. The accuracy of model $\theta_A$ is 1.0. The accuracy of only gradient $g$ is 0.72.

| Dataset | CIFAR10 | | | | |
|---|---|---|---|---|---|
| Scale factor | $\theta_1 + \alpha g$ | $\theta_2 + \alpha g$ | $\theta_3 + \alpha g$ | $\theta_4 + \alpha g$ | $\theta_5 + \alpha g$ |
| $\alpha$=0.01 | 0.10 | 0.10 | 0.10 | 0.10 | 0.10 |
| $\alpha$=0.1 | 0.10 | 0.10 | 0.10 | 0.07 | 0.10 |
| $\alpha$=0.2 | 0.10 | 0.07 | 0.05 | 0.07 | 0.10 |
| $\alpha$=0.5 | 0.12 | 0.17 | 0.07 | 0.20 | 0.25 |
| $\alpha$=0.8 | 0.17 | 0.30 | 0.27 | 0.32 | 0.32 |
| $\alpha$=1.0 | 0.35 | 0.35 | 0.42 | 0.35 | 0.40 |
| $\alpha$=5.0 | 0.75 | 0.72 | 0.72 | 0.77 | 0.72 |
| $\alpha$=10 | 0.77 | **0.80** | 0.72 | 0.77 | 0.75 |
| $\alpha$=20 | 0.77 | **0.80** | 0.77 | 0.77 | 0.75 |
| $\alpha$=30 | 0.77 | **0.80** | 0.77 | 0.77 | 0.77 |
| $\alpha$=40 | 0.77 | **0.80** | 0.77 | 0.77 | 0.77 |

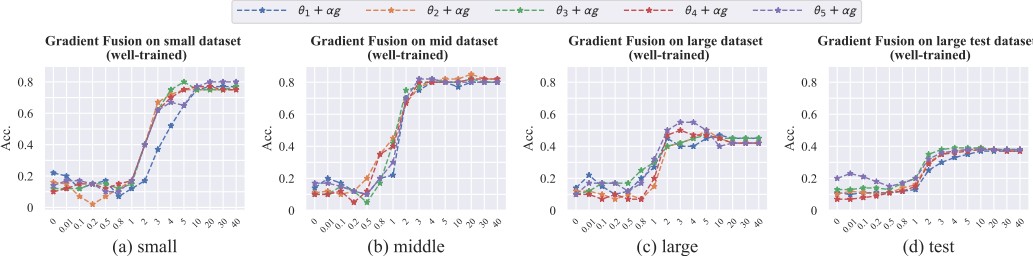

Figure 2: Empirical results for gradient assignment with different well-trained models. (a), (b), (c), (d) respectively show the results obtained on the small, medium, and large-scale models and the results on the test set. $\theta_1$ through $\theta_5$ respectively denote the parameters of the trained model on five different datasets: CIFAR100 Krizhevsky (2009), Caltech256 Griffin et al. (2022a), Food-101 Bossard et al. (2014), MNIST LeCun et al. (1998), and STL-10 Coates et al. (2011).

initial points $\theta_i$ and gradients $g$. However, none of them achieves an accuracy close to 1.0, as observed in the original model. This suggests that assigning cumulative gradients leads to some representation loss. This could be attributed to two reasons. (1) Although $\theta_0$ is randomly initialized, it contains some basic structures that can accurately represent some of the training samples, achieving approximately 0.1 accuracy for this 10-way classification problem. These structures do not need to be altered by the cumulative gradients, so the cumulative gradients $g$ will not contain the representations from such basic structures. This results in some performance loss when assigned to other models. (2) The randomly initialized part in the weights of other models $\theta_i$ has strong cancellation structures with the cumulative gradients $g$ Li & Liang (2018), which could also result in some performance loss.

Similar to the results on randomly initialized points, we observe an increase in accuracy when adding the cumulative gradient $g$ to well-trained models with the rise of $\alpha$. These results are depicted in Figure 2. Remarkably, the new models $\theta_n$ at the medium-scale are able to achieve accuracies exceeding 0.8, despite not being trained on CIFAR10. Furthermore, the gradient-assigned models also exhibit promising performance on the test set, as depicted in Figure 2(d) and Table 2.

Comparing the first three subfigures, we observe the accuracies at the large scale are significantly lower than those at the small and medium scales. This is expected as the model trained on the larger dataset is more complex, increasing the likelihood of information loss during the transfer process. However, upon closer inspection, we note that the gap between the curves becomes negligible when $\alpha$ is sufficiently large (greater than 10). This suggests that as the proportion of gradient $g$ increases, the impact caused by the difference in $\theta_i$ diminishes.

Table 2: Putting the large-scale gradient-assigned model on test set, where $\theta_i$ is well-trained model. The accuracy of model $\theta_A$ on test set is 0.61. The accuracy of only gradient $g$ on test set is 0.40.

| Dataset | CIFAR10 | | | | |
|---|---|---|---|---|---|
| Scale factor | $\theta_1 + \alpha g$ | $\theta_2 + \alpha g$ | $\theta_3 + \alpha g$ | $\theta_4 + \alpha g$ | $\theta_5 + \alpha g$ |
| $\alpha$=0.01 | 0.10 | 0.12 | 0.13 | 0.07 | 0.23 |
| $\alpha$=0.1 | 0.11 | 0.11 | 0.14 | 0.08 | 0.21 |
| $\alpha$=0.2 | 0.11 | 0.10 | 0.14 | 0.09 | 0.18 |
| $\alpha$=0.5 | 0.11 | 0.11 | 0.13 | 0.11 | 0.15 |
| $\alpha$=0.8 | 0.12 | 0.14 | 0.17 | 0.12 | 0.17 |
| $\alpha$=1.0 | 0.13 | 0.16 | 0.20 | 0.15 | 0.20 |
| $\alpha$=2.0 | 0.25 | 0.30 | 0.35 | 0.29 | 0.32 |
| $\alpha$=3.0 | 0.30 | 0.35 | 0.38 | 0.35 | 0.36 |
| $\alpha$=4.0 | 0.33 | 0.37 | **0.39** | 0.36 | 0.37 |
| $\alpha$=5.0 | 0.35 | 0.38 | **0.39** | 0.37 | 0.38 |
| $\alpha$=10 | 0.37 | **0.39** | **0.39** | 0.38 | 0.38 |
| $\alpha$=20 | 0.37 | 0.38 | 0.38 | 0.38 | 0.38 |
| $\alpha$=30 | 0.37 | 0.38 | 0.38 | 0.38 | 0.38 |
| $\alpha$=40 | 0.37 | 0.38 | 0.38 | 0.37 | 0.38 |

Given these empirical results, we can conclude that the cumulative gradients $g$ can be assigned to a new model $\theta_n$, regardless of whether the model is randomly initialized or well-trained, to facilitate knowledge transfer from model $\theta_A$ to $\theta_n$. Interestingly, the representations of the cumulative gradients $g$ only partially depend on the corresponding initial point. Despite being counterintuitive, this phenomenon offers a new perspective on the knowledge transfer process in neural networks.

## 4 MODEL FUSION

### 4.1 ASSIGNING WEIGHTS OF A LEARNED MODEL TO DIFFERENT MODELS

In this section, we introduce the concept of "model fusion", which extends the idea of assigning accumulative gradients to the assignment of model weights. As we discussed in Section 3, our empirical results demonstrate that the accumulative gradients $g$ can be assigned to a fresh model $\theta_n$, no matter it is randomly initialized or well-trained, to facilitate knowledge transfer from model $\theta_A$ to $\theta_n$. In fact, we can consider the well-trained model $\theta_A$, which encompassing the knowledge embedded in $g$, as accumulative gradients that can be assigned to a new model to achieve knowledge transfer. Hence, analogous to assigning accumulative gradients, we can directly assign the learned weights $\theta_A$ to a new model $\theta_n$ as follows:

$$\tilde{\theta}_n = \theta_n + \alpha \theta_A. \tag{7}$$

In alignment with Section 3, we executed experiments on several randomly initialized models and well-trained models, all denoted as $\theta_n$. The models used as $\theta_A$ were trained on the CIFAR10 dataset. We conducted experiments at three distinct scales: small, medium, and large, with settings and network structures consistent with those outlined in Section 3. The five well-trained models were trained on various datasets, including CIFAR100, Caltech256, Food-101, MNIST, and STL-10.

### 4.2 EMPIRICAL RESULTS FOR MODEL FUSION

The experiment results are presented in Figure 3. The accuracy curves in all four subfigures exhibit an increasing trend with increasing $\alpha$. Surprisingly, the fused models achieve accuracies of 0.8 and above on CIFAR10, even with accuracies close to 1.0 (e.g., 0.93) being achieved under small and medium scales. The corresponding experimental values of Figure 3(a) are summarized in Table 3.

In this table, the test accuracy of the fusion model on CIFAR10 significantly improves when the value of $\alpha$ is increased to 0.4. With the increase of $\alpha$, the accuracy can reach up to 0.93, as shown in the third column of the penultimate row of Table 3. Although the highest accuracy achieved in the

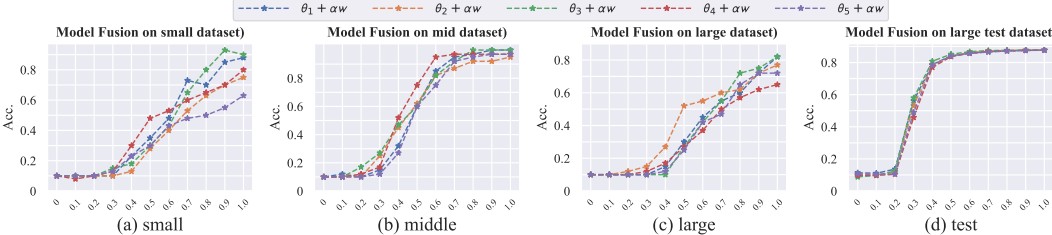

Figure 3: Empirical results for model fusion with random initialized parameters. $\theta_1$ through $\theta_5$ are five different model parameters. (a), (b) and (c) respectively show the results on small, medium and large scales. (d) shows the results on test set.

Table 3: Assigning the learned weights $\theta_A$, updated by small-scale accumulative gradients, to random initialized models, where $\theta_i$ is random initialized model. The test accuracy of model $\theta_A$ is 1.0.

| Dataset | CIFAR10 | | | | |
|---|---|---|---|---|---|
| Scale factor | $\theta_1 + \alpha\theta_A$ | $\theta_2 + \alpha\theta_A$ | $\theta_3 + \alpha\theta_A$ | $\theta_4 + \alpha\theta_A$ | $\theta_5 + \alpha\theta_A$ |
| $\alpha$=0.1 | 0.10 | 0.10 | 0.10 | 0.08 | 0.10 |
| $\alpha$=0.2 | 0.10 | 0.10 | 0.10 | 0.10 | 0.10 |
| $\alpha$=0.3 | 0.10 | 0.10 | 0.15 | 0.13 | 0.13 |
| $\alpha$=0.4 | 0.23 | 0.13 | 0.18 | 0.30 | 0.23 |
| $\alpha$=0.5 | 0.35 | 0.28 | 0.30 | 0.48 | 0.30 |
| $\alpha$=0.6 | 0.48 | 0.40 | 0.43 | 0.53 | 0.43 |
| $\alpha$=0.7 | 0.73 | 0.53 | 0.65 | 0.60 | 0.48 |
| $\alpha$=0.8 | 0.70 | 0.63 | 0.80 | 0.65 | 0.50 |
| $\alpha$=0.9 | 0.85 | 0.70 | **0.93** | 0.70 | 0.55 |
| $\alpha$=1.0 | 0.88 | 0.75 | 0.90 | 0.80 | 0.63 |

other columns is not more than 0.9, an accuracy of 0.88 is still remarkable for an untrained fusion model. These results present a significant improvement compared with the results in Section 3, where only the gradients are added to the model. It indicates that the training information contained in the weights is more comprehensive compared with the gradient.

The experimental results with well-trained models also demonstrate similar outcomes. Figure 4 presents the accuracy under small and medium scale can exceed 0.95, which can mostly recover the original accuracy of $\theta_A$. The accuracies of fusion with each $\theta_n$ at the large-scale experiments are also greater than 0.6, which is a considerable improvement compared with the results of the large scale in Section 3.2. Furthermore, the fusion model also exhibits promising performance on the test set, as depicted in Figure 4(d) and Table 4. The results suggest that weights $\theta_A$ offer better guidance to the model than gradients $g$ in classifying the CIFAR10 dataset.

Another noteworthy point is that comparing Figure 1 and Figure 3, the accuracy of the fusion models grows faster than the accuracy of the gradient-assigned models. The model obtained by one-to-one fusion with the randomly initialized $\theta_n$ does not work well when only the gradients are assigned (even with poor results, see the small-scale results represented in subfigure (a)). Accuracy improves only when $\alpha$ becomes larger, e.g., to 5. However, summing with weights, $\alpha$=0.8 gives a high accuracy. This again supports the view that the information contained in the model weights is more comprehensive than that contained in the gradients and thus can affect $\theta_n$ faster and more significantly. This validates the first reason discussed in Section 3.2. More precisely, the randomly initialized $\theta_0$ contained some basic and useful structures for the given task $A$, and the stochastic gradient optimization method does not change this structure. A learned neural network representation consists of accumulative gradients and such structure.

The results presented above demonstrate that the learned weights $\theta_A$ can be assigned to a new model $\theta_n$ using a simple summation operation, just like accumulative gradients. This operation can effectively transfer the dataset information contained in $\theta_A$ to the new weights, resulting in good

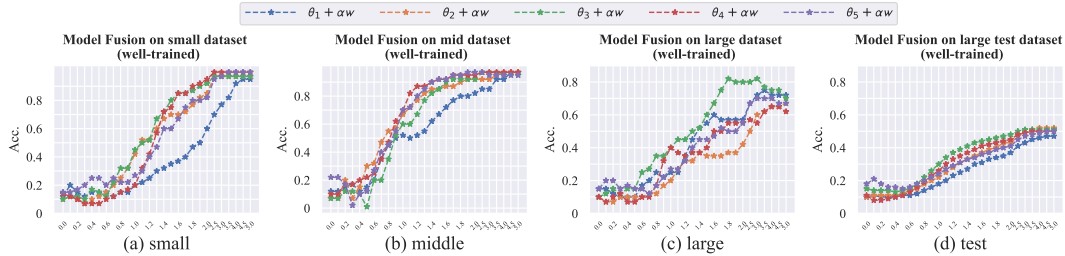

Figure 4: Empirical results for model fusion with different well-trained models. $\theta_1 \sim \theta_5$ represent the same content as Figure 2, which are the trained models on the CIFAR100, Caltech256, Food-101, MNIST, and STL-10 datasets.

Table 4: Putting the large-scale fusion model on test set, where $\theta_i$ is well-trained model. The accuracy of model $\theta_A$ on test set is 0.61.

| Dataset | CIFAR10 | | | | |
|---|---|---|---|---|---|
| Scale factor | $\theta_1 + \alpha\theta_A$ | $\theta_2 + \alpha\theta_A$ | $\theta_3 + \alpha\theta_A$ | $\theta_4 + \alpha\theta_A$ | $\theta_5 + \alpha\theta_A$ |
| $\alpha$=0.1 | 0.11 | 0.11 | 0.14 | 0.08 | 0.21 |
| $\alpha$=0.5 | 0.11 | 0.12 | 0.14 | 0.11 | 0.15 |
| $\alpha$=1.0 | 0.18 | 0.22 | 0.30 | 0.26 | 0.24 |
| $\alpha$=1.5 | 0.30 | 0.35 | 0.43 | 0.39 | 0.34 |
| $\alpha$=2.0 | 0.37 | 0.43 | 0.48 | 0.45 | 0.41 |
| $\alpha$=3.0 | 0.43 | 0.49 | 0.51 | 0.49 | 0.47 |
| $\alpha$=4.0 | 0.46 | **0.52** | 0.51 | 0.50 | 0.49 |
| $\alpha$=5.0 | 0.47 | **0.52** | 0.51 | 0.50 | 0.50 |

performance on the task at hand. Furthermore, our experiments show that the learned weights $\theta_A$ contain basic structures for good representations that the accumulative gradients $g$ may not capture. This suggests that model fusion can lead to much better knowledge transfer from model $\theta_A$ to $\theta_n$.

## 5 DEALING WITH CATASTROPHIC FORGETTING

### 5.1 DIFFERENCE BETWEEN GRADIENT ASSIGNMENT AND NORMAL TRAINING

Recent studies suggest that the mammalian brain can reduce the plasticity of synapses, which is critical to previously learned tasks, to avoid catastrophic forgetting. This strategy is also useful for neural networks, and Kirkpatrick et al. Kirkpatrick et al. (2017) implemented an elastic weight consolidation algorithm by constraining neural network parameters to stay close to the previously learned parameters. However, such constraint can limit the learning process of new tasks and lead to lower performance. Based on the empirical results of Section 3 and Section 4, we can assign accumulative gradients $g$ or the parameters of a model $\theta_A$ to another new model $\theta_n$. In fact, if the new model $\theta_n$ is trained on other tasks, e.g., task $B$, the new model $\tilde{\theta}_n$ may have a chance to maintain knowledge from both tasks $A$ and $B$ simultaneously.

Why assigning accumulative gradients and model fusion can alleviate catastrophic forgetting? To answer this question, we first formalize the derivations to demonstrate the difference between gradient assignment and the normal training process, training task $B$ first and task $A$ later. Then, we employ Taylor series expansion to approximate those two processes to demonstrate how the cancellation structure leads the model to catastrophic forgetting. Finally, we provide empirical results to verify this theoretic analysis and draw a conclusion that assigning accumulative gradients and model fusion can alleviate catastrophic forgetting. For clarity, we put the full derivation and analysis process in Section B.1 in Appendix.

Table 5: Results of traditional training methods.

| Dataset Model | Training set | | Test set | |
|---|---|---|---|---|
| | CIFAR10 | CIFAR100 | CIFAR10 | CIFAR100 |
| CIFAR10->100 | 25 | 100 | 23 | 88 |
| CIFAR100->10 | 100 | 9 | 88 | 11 |

Table 6: The **fine-grained** results of putting the fusion model trained by **large-scale** on **testing** set of **CIFAR10**.

| $\alpha\theta_A + \beta\theta_B$ | CIFAR10-test-finegrained | | | | | | | | | |
|---|---|---|---|---|---|---|---|---|---|---|
| $\beta$ \ $\alpha$ | 0.50 | 0.60 | 0.70 | 0.80 | 0.90 | 1.00 | 2.00 | 3.00 | 4.00 | 5.00 |
| 0.50 | 79.74 | 82.61 | 83.99 | 84.92 | 85.48 | 85.80 | 86.98 | 87.15 | 87.23 | 87.17 |
| 0.60 | 74.90 | 79.73 | 82.44 | 83.57 | 84.49 | 85.13 | 86.82 | 87.07 | 87.15 | 87.22 |
| 0.70 | 67.52 | 75.65 | 79.77 | 82.21 | 83.29 | 84.19 | 86.72 | 87.01 | 87.12 | 87.18 |
| 0.80 | 57.83 | 70.42 | 76.40 | 79.78 | 81.82 | 82.97 | 86.42 | 86.92 | 87.08 | 87.18 |
| 0.90 | 47.74 | 62.97 | 72.14 | 76.94 | 79.77 | 81.67 | 86.10 | 86.82 | 87.04 | 87.11 |
| 1.00 | 37.99 | 54.36 | 66.39 | 73.39 | 77.27 | 79.77 | 85.82 | 86.77 | 86.99 | 87.09 |
| 2.00 | 14.31 | 16.75 | 19.66 | 23.99 | 30.37 | 38.03 | 79.74 | 84.52 | 85.84 | 86.44 |
| 3.00 | 12.04 | 12.87 | 13.76 | 14.98 | 16.73 | 18.71 | **63.08** | 79.76 | 83.62 | 85.14 |
| 4.00 | 11.42 | 11.83 | 12.21 | 12.86 | 13.50 | 14.35 | 38.04 | 70.47 | 79.76 | 82.94 |
| 5.00 | 11.23 | 11.39 | 11.70 | 12.00 | 12.30 | 12.87 | 24.02 | 54.42 | 73.46 | 79.77 |

## 5.2 EMPIRICAL RESULTS FOR CATASTROPHIC FORGETTING

To demonstrate the effectiveness of our method in addressing the catastrophic forgetting problem, we first sequentially train on the CIFAR10 and CIFAR100 datasets using the traditional method and record the test results. As shown in Table 5, CIFAR10->100 indicates that the initialized model parameter $\theta_0$ is first trained on CIFAR10 and then used for training on CIFAR100. CIFAR100->10 means the reverse order.

The values in the table indicate that the model performs significantly better on the last trained dataset compared with the previous one. This suggests that the model tends to remember information only from the last training dataset while forgetting most of the information learned from previous training experiences.

In contrast, the experimental results of our proposed fusion model on these two datasets are presented in Table 6. More detailed results can be found in Section B.3 in Appendix. Experimentally, our method has demonstrated the ability to alleviate this phenomenon to some extent. In this table, $\theta_A$ represents the well-trained model on the CIFAR10 dataset, and $\theta_B$ represents the model trained on the CIFAR100 dataset. Both models are initialized with $\theta_0$. The coefficients $\alpha$ and $\beta$ are used for weight fusion. The accuracies in the table are reported on a hundred-point scale.

## 6 CONCLUSION AND FUTURE WORK

In this work, we presented a simple yet intriguing phenomenon, which sheds light on the black box nature of deep neural networks and their associated gradients. Through extensive experiments, we have demonstrated that the representation captured by accumulative gradients is not solely reliant on the initial point from which these gradients are computed. Consequently, these accumulated gradients can be effectively assigned to different models, enabling the transfer of learned representations.

We have also observed that the weights of a neural network can also be regarded as gradients and assigned to different models for knowledge transfer. Additionally, our experiments have revealed that simply scaling the weights or accumulated gradients has minimal impact on the overall representation. These results indicate the existence of numerous unexplored black boxes within deep neural networks, leaving ample room for future investigation.

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

# A EXPERIMENTS SETTING

## A.1 DATASETS

We are primarily concerned with image classification datasets, where model A is trained on the CIFAR10 dataset. The other five datasets, namely CIFAR100, Caltech256, Food101, MNIST and STL-10, are used as $\theta_0^i$. The experiments involve various classes, end the descriptions of these classes are provided below. The images used for each class are randomly selected.

**CIFAR10:** The CIFAR-10 datasetKrizhevsky (2009) comprises 60,000 32x32 color images divided into 10 classes, with 6,000 images in each category. The dataset is further split into a training set of 50,000 images and a test set of 10,000 images. The test set consists of precisely 1,000 images randomly selected from each class, while the training set contains exactly 5,000 images from each class. The CIFAR-10 dataset includes the following classes: airplane, automobile, bird, cat, deer, dog, frog, horse, ship, and truck, a total of 10 classes.

**CIFAR100:** The CIFAR100 datasetKrizhevsky (2009) is similar to CIFAR-10, but it consists of 100 classes with 600 images per class. Each category in CIFAR-100 contains 500 training images and 100 testing images. For our experiments, we have chosen 10 specific classes: seal, man, palm-tree, woman, wardrobe, beaver, plain, pickup-truck, skyscraper, and possum.

**Caltech256:** The Caltech-256 datasetGriffin et al. (2022b) is a widely used object recognition dataset that consists of 30,607 real-world images of various sizes. It includes a total of 257 classes (256 object classes and an additional clutter class). Each class is represented by a mininum of 80 images. For our experiments, we select 10 specific classes from the Caltech-256 dataset: beer-mug, car-tire, hummingbird, killer-whale, knife, mushroom, scorpion-101, speed-boat, teapot, and tomato.

**Food101:** The Food101 datasetBossard et al. (2014) is a popular dataset for food classification, containing 101 food classes with 101,000 images. Each class includes 250 human-censored test images and 750 training images. In our experiments, we have focused on 10 specific classes from Food101 dataset: spaghetti-bolognese, carrot-cake, pad-thai, french-toast, garlic-bread, tiramisu, creme-brulee, samosa, chocolate-mousse, and poutine.

**MINIST:** The MINIST datasetLeCun et al. (1998) is a widely used dataset for handwritten digit recognition. It consists of 70,000 grayscale images of handwritten digits with a resolution of 28x28 pixels. The dataset is divided into a training set of 60,000 examples and a test set of 10,000 examples. The digits in the images have been size normalized and centered within the fixed-size image. The dataset includes '0', '1', '2', '3', '4', '5', '6', '7', '8', and '9', making a total of 10 classes.

**STL-10:** The STL-10 datasetCoates et al. (2011) is a well-known image recognition dataset used for various tasks in machine learning. It contains a total of 10 classes, includingairplane, bird, car, cat, deer, dog, horse, monkey, ship, and truck. The dataset consists of 500 training images and 800 testing images per class, amounting to a total of 5,000 training images and 8,000 testing images. All images in the dataset have a resolution of 96x96 pixels and are in color.

## A.2 ARCHITECTURE OF THE NETWORK

To ensure the generalizability of the experimental results and minimize the impact of neural network architecture, we perform all experiments using simple and widely adopted network structures. The details of these architectures are provided below.

**Assigning Accumulative Gradients:** In this set of experiments, we employ a standard convolution neural network (CNN) architecture. The images are processed through a sequence of two convolution layers and three fully-connected layers, with the Rectified Linear Unit (Relu) activation function applied after each layer. Both convolutional layers have a kernel size of 5, and the number of output channels is set to 6 and 16 respectively. Following each convolutional layer, a maximum pooling layer is applied to downsample the feature maps.

**Model Fusion:** Same as the architecture described in the previous section (Assigning Accumulative Gradients).

**Catastrophic Forgetting:** In this part of the experiments, we utilize residual structures proposed by He et al.He et al. (2016). The architecture includes multiple convolution blocks, each containing a

convolutional layer, a Batch Normalization (BN) layerIoffe & Szegedy (2015), and a ReLU activation layer. The images are processed through a total of 8 convolution blocks. To introduce shortcut connections and enable easier flow of gradients during training, a shortcut connection is inserted between the output ends of the second and fourth convolution blocks. Similarly, another shortcut connection is added after the sixth and eighth convolution blocks. Finally, the output of the last convolution block is passed through a classifier, which consists of a maximum pooling layer for downsampling and a linear layer for the final classification.

# B    DEALING WITH CATASTROPHIC FORGETTING

## B.1    THE DIFFERENCE BETWEEN GRADIENTS ASSIGNMENT AND NORMAL TRAINING

Recent studies have highlighted the phenomenon of reduced synapse plasticity in the mammalian brain, which serves as a mechanism to prevent catastrophic forgetting of previously learned tasks. This adaptive strategy has also been explored in neural networks. Kirkpatrick et al. **?** implemented the Elastic weight consolidation algorithm, which constrains the parameters of a neural network to remain close to the parameters learned from previous tasks. However, such constraints can impede the learning process of new tasks and result in decreased performance. Building upon the empirical findings presented in the previous sections (Section 3 and Section 4 in the main text), we propose an alternative approach to address catastrophic forgetting. Specifically, we investigate the assignment of accumulative gradients $g$ or the parameters of a well-trained model $\theta_A$ to a new model $\theta_n$. By employing this strategy, if the new model $\theta_n$ is trained on a different task, such as task B, there is a possibility for the new model, denoted as $\tilde{\theta}_n$, to retain knowledge from both task A and task B simultaneously.

The underlying question of why assigning accumulative gradients and utilizing model fusion can mitigate catastrophic forgetting is of paramount importance. To address this inquiry, we will commence by formalizing the derivations to illustrate the distinction between gradient assignment and the conventional training procedure, where task B is trained before task A. Subsequently, we will employ the Taylor series expansion to approximate these two processes, shedding light on how the cancellation structure within them contributes to catastrophic forgetting. To further substantiate our theoretical analysis, we will present empirical results that serve to validate the proposed approach. Through comprehensive experiments, we will demonstrate the efficacy of assigning accumulative gradients and model fusion in ameliorating catastrophic forgetting.

Formally, given two tasks, A and B, with the corresponding datasets $\mathcal{D}_A$ and $\mathcal{D}_B$, respectively. For a random initialized model $\theta_0$, we first train it with task B to get a well-trained model $\theta_B$:

$$\theta_B = \theta_0 - \beta_B \nabla_{\theta_0} \sum_{(x,y)\in\mathcal{D}_B} \mathcal{L}_B,$$

where $\beta_B$ is the learning rate in task B, $\mathcal{L}_B$ is the loss function of task B. Then, we train task A with the well-trained model $\theta_B$. Thus, the normal training process $\theta_{BA}$ can be formulated as:

$$\theta_{BA} = \theta_B - \beta_A \nabla_{\theta_B} \sum_{(x,y)\in\mathcal{D}_A} \mathcal{L}_A$$

$$= \theta_0 - \beta_B \nabla_{\theta_0} \sum_{(x,y)\in\mathcal{D}_B} \mathcal{L}_B - \beta_A \nabla_{\theta_B} \sum_{(x,y)\in\mathcal{D}_A} \mathcal{L}_A,$$

where $\beta_A$ is the learning rate in task A, $\mathcal{L}_A$ is the loss function of task A.

In contrast, the process of assigning accumulative gradients $\theta'_{BA}$ can be formulated as:

$$\theta'_{BA} = \theta_B + \alpha g$$

$$= \theta_B - \alpha \beta_A \nabla_{\theta_0} \sum_{(x,y)\in\mathcal{D}_A} \mathcal{L}_A$$

$$= \theta_0 - \beta_B \nabla_{\theta_0} \sum_{(x,y)\in\mathcal{D}_B} \mathcal{L}_B - \alpha \beta \nabla_{\theta_0} \sum_{(x,y)\in\mathcal{D}_A} \mathcal{L}_A$$

$$= \theta_0 - \beta_B \nabla_{\theta_0} \sum_{(x,y)\in\mathcal{D}_B} \mathcal{L}_B - \beta_A \nabla_{\theta_0} \sum_{(x,y)\in\mathcal{D}_A} \mathcal{L}_A \ \ (\alpha = 1).$$

Note that, for simple and clear, we assume both task A and task B start from the same initial point $\theta_0$, though different initial point for task A and task B can also obtain similar results. Note that if we set $\alpha = 1$, the difference between normal training process $\theta_{AB}$ and assigning accumulative gradients $\theta'_{AB}$ are the terms $g_{AB} = \beta_A \nabla_{\theta_B} \sum_{(x,y) \in \mathcal{D}_A} \mathcal{L}_A$ and $g'_{AB} = \beta_A \nabla_{\theta_0} \sum_{(x,y) \in \mathcal{D}_A} \mathcal{L}_A$. In term $\beta_A \nabla_{\theta_B} \sum_{(x,y) \in \mathcal{D}_A} \mathcal{L}_A$ of normal training process, model $\theta_{BA}$ is updated from $\theta_B$. In contrast, in term $\beta_A \nabla_{\theta_0} \sum_{(x,y) \in \mathcal{D}_A} \mathcal{L}_A$ of assigning accumulative gradients, model $\theta'_{BA}$ is updated from $\theta_0$. Such difference leads to different results for $\theta_{BA}$ and $\theta'_{BA}$ over task A and task B. Specifically, $\theta_{BA}$ almost forgets the learned knowledge of task B, but $\theta'_{BA}$ can still hold some useful structure to represent the learned knowledge of task B. Next, we will analyze what leads to these differences.

## B.2 How is the cancellation structure leading to catastrophic forgetting

In this section, we will introduce the Taylor series expansion as a means to reframe the process of assigning accumulative gradients and the normal training process into two forms of cancellation terms. By employing this expansion, we can gain a comprehensive understanding of the factors that contribute to the degradation of the representation of task B during the training process of task A. This deeper understanding enables us to point out that instead of the conventional one-by-one training process, combining and leveraging the gradients obtained from multiple tasks can effectively mitigate catastrophic forgetting and enhance the overall performance of the model. We will use the following definitions:

$$\begin{aligned}
g_i &= \mathcal{L}'_i(\theta_{i-1}), (gradient\ of\ corresponding\ loss) \\
H_i &= \mathcal{L}''_i(\theta_{i-1}), (Hessian\ of\ corresponding\ loss) \\
\theta_i &= \theta_{i-1} - \alpha g_i, (parameter\ vectors) \\
\hat{g}_A &= \mathcal{L}'_A(\theta_0), (gradient\ at\ the\ initial\ point) \\
\hat{H}_A &= \mathcal{L}''_A(\theta_0), (Hessian\ at\ the\ initial\ point) \\
\hat{g}_B &= \mathcal{L}'_B(\theta_0), (gradient\ at\ the\ initial\ point) \\
\hat{H}_B &= \mathcal{L}''_B(\theta_0), (Hessian\ at\ the\ initial\ point)
\end{aligned}$$

where $i \in [0, A, B]$. Next, let's approximate the SGD gradient term $g_{AB} = \nabla_{\theta_A} \sum_{(x,y) \in \mathcal{D}_B} \mathcal{L}_B$ of normal training process to the initial point $\theta_0$ to $O\left(\theta_B^2\right)$ as follow:

$$\begin{aligned}
g_{AB} &= \mathcal{L}'_A(\theta_B) = \mathcal{L}'_A(\theta_0) + \mathcal{L}''_A(\theta_0)(\theta_B - \theta_0) \\
&+ \underbrace{O(\|\theta_B - \theta_0\|^2)}_{=O(\beta_B^2)} \quad \text{(Taylor's theorem)} \\
&= \hat{g}_A + \hat{H}_A(\theta_B - \theta_0) + O(\beta_B^2) \\
&\quad \text{(using definition of } \hat{g}_i, \hat{H}_i) \\
&= \hat{g}_A - \beta_B \hat{H}_A \hat{g}_B + O(\beta_B^2), \\
&\quad \text{(using } \theta_B - \theta_0 = -\beta_B \hat{g}_B)
\end{aligned}$$

Then, we can reformulate the SGD gradient term $g'_{AB} = \beta_A \nabla_{\theta_0} \sum_{(x,y) \in \mathcal{D}_A} \mathcal{L}_A$ of assigning accumulative gradients as follow:

$$g'_{AB} = \beta_A \nabla_{\theta_0} \sum_{(x,y) \in \mathcal{D}_A} = \hat{g}_A.$$

That is to say, if we ignore the infinitesimal term $O\left(\theta_B^2\right)$, the difference between $\theta_{AB}$ and $\theta_{AB}$ is the only term $-\beta_B \hat{H}_A \hat{g}_B$. So, the question is, what is the effect of such a term and what is its relationship to catastrophic forgetting?

To answer such a question, we first discuss the effect of such a term in the same task. As discussed in Nichol et al. (2018), if the gradients $\hat{H}$ and $\hat{g}$ comes from different mini-batches of a given task $T$,

the term $\hat{H}_T \hat{g}_T$ can improve generalization of model. In addition, as stated in Lu et al. (2020), the gradients come from different samples and can be treated as regularization on the model. Thus, in a given task $T$, the effect of the term $\hat{H}_T \hat{g}_T$ is to regularize the representation of $\hat{g}_T$ of a given model $\theta_T = \theta_0 + \alpha \hat{g}_T$ to improve generalization.

Similarly, in multi-task settings, e.g., task A and task B, the gradients that come from different tasks can also be treated as regularization on the model. However, unlike gradients, $g_T$ come from one task can lead the model to the same optimal (or local optimal), $\hat{g}_A$ and $\hat{g}_B$ lead to different points, which are optimal (or local optimal) for task A and task B, respectively. As a consequence, the effect of the term $-\beta_B \hat{H}_A \hat{g}_B$ is to regularize the representation of $\hat{g}_B$ of a given model $\theta_B = \theta_0 + \alpha \hat{g}_B$ to the optimal (or local optimal) of task A and improve generalization over task A. In other words, the learned structure of task B of model $\theta_B$ will be destroyed (the representation of $\hat{g}_B$ has been regularized) by the newly updating via the gradient $\hat{g}_A$ from task A. Finally, the well-trained model $\theta_{AB}$ will only hold the representation of task A and can not represent task B. As a result, the catastrophic forgetting problem happens.

We name such critical term $-\beta_B \hat{H}_A \hat{g}_B$ as the Gradient of Gradient Cancellation (GGC) term. With extra GGC term, $g_{AB}$ will lead more significant performance decline over task B than $g_{AB}\prime$. Due to the representation of $\hat{g}_B$ has been destroyed. However, the representation of $\hat{g}_B$ is not only affected by the term GGC but also by the conflicting gradients Liu et al. (2021) between $\hat{g}_B$ and $\hat{g}_A$. When this conflict is large, the following gradients will decrease the performance on the former learned knowledge.

### B.3 EMPIRICAL RESULTS FOR CATASTROPHIC FORGETTING

To demonstrate the effectiveness of our method in addressing the catastrophic forgetting problem, we conducted sequential training on the CIFAR10 and CIFAR100 using the traditional approach. The test results are presented in Table 5 in the main text. In this table, CIFAR10->100 denotes the scenario where the model parameter $\theta_0$ is initially trained on CIFAR10 and subsequently fine-tuned on CIFAR100. Conversely, CIFAR100->10 refers to the reverse order of training.

The results presented in the table clearly indicate that the performance of the model sinificantly improves on the last trained dataset compared to the previous one. These findings suggest that the model tends to primarily retain the information from the most recent training dataset while forgetting a substantial portion of the knowledge acquired in the previous training experiences.

The finegrained experimental results of the proposed fusion model on the CIFAR10 and CIFAR100 datasets are presented in Table 6 (see the main text) and Table 1 (see the Appendix). These tables illustrate the impact of our method on alleviating the phenomenon of catastrophic forgetting.

In the tables, $\theta_A$ represents the well-trained model on the CIFAR10 dataset, while $\theta_B$ represents the well-trained model on the CIFAR100 dataset. Both models are initialized with $\theta_0$. The coefficients of weights fusion, denoted as $\alpha$ and $\beta$, play a crucial role in our fusion model. The accuracies reported in the tables are based on a hundred-point system.

The accuracy of the model on CIFAR10 is higher when the coefficient $\alpha$ is larger and $\beta$ is smaller. Conversely, when $\alpha$ is smaller and $\beta$ is larger, the model achieves higher accuracy on CIFAR100. Therefore, the values in the two tables exhibit precisely opposite trends. On CIFAR10, the accuracies gradually increase towards the upper right corner of the table, whereas on CIFAR100, high accuracies are concentrated in the lower left corner. Notably, we have identified a set of values near the diagonal of the table where their corresponding $\alpha$ and $\beta$ yield a fusion model that performs well on both datasets. This set of values has been highlighted in bold in the table. Specifically, when $\alpha = 2$ and $\beta = 3$, the fusion model achieves an accuracy exceeding 0.6 on the test sets for both CIFAR10 and CIFAR100. This remarkable finding demonstrates that, compared to the traditional approach outlined in Table 5 (see the main text), our proposed method successfully mitigates catastrophic forgetting by identifying a balanced model that effectively retains information from both datasets.

### B.3.1 DETAILED RESULTS FOR GRADIENTS ASSIGNMENT

The present section provides a comprehensive overview of the empirical results obtained from the accumulative gradients experiments. These experiments encompass small, medium, and large-scale

Table 1: The **finegrained** results of putting the fusion model trained by **large-scale** on **testing** set of **CIFAR100**.

| $\alpha\theta_A + \beta\theta_B$ | CIFAR100-test-finegrained | | | | | | | | | |
|---|---|---|---|---|---|---|---|---|---|---|
| $\beta$ \ $\alpha$ | 0.50 | 0.60 | 0.70 | 0.80 | 0.90 | 1.00 | 2.00 | 3.00 | 4.00 | 5.00 |
| 0.50 | 29.90 | 23.10 | 20.10 | 17.50 | 15.70 | 14.70 | 11.40 | 10.20 | 9.80 | 9.70 |
| 0.60 | 42.00 | 29.90 | 24.20 | 21.10 | 18.90 | 16.70 | 12.50 | 10.80 | 10.20 | 9.80 |
| 0.70 | 54.20 | 39.00 | 29.90 | 24.60 | 21.50 | 20.00 | 13.30 | 11.10 | 10.20 | 10.10 |
| 0.80 | 68.70 | 50.80 | 37.30 | 29.90 | 25.10 | 22.20 | 13.70 | 11.80 | 10.80 | 10.30 |
| 0.90 | 74.40 | 61.80 | 47.30 | 36.70 | 29.70 | 25.60 | 14.00 | 12.60 | 11.00 | 10.30 |
| 1.00 | 77.20 | 70.50 | 57.10 | 45.00 | 35.90 | 29.70 | 14.50 | 12.90 | 11.50 | 10.80 |
| 2.00 | 85.60 | 84.80 | 84.40 | 82.80 | 80.60 | 77.10 | 29.70 | 19.00 | 14.50 | 13.80 |
| 3.00 | 86.60 | 86.30 | 85.70 | 85.20 | 84.80 | 84.40 | **62.20** | 29.70 | 21.30 | 17.00 |
| 4.00 | 87.10 | 86.70 | 86.30 | 86.30 | 85.90 | 85.60 | 77.10 | 51.00 | 29.70 | 22.10 |
| 5.00 | 87.00 | 87.10 | 86.80 | 86.70 | 86.20 | 86.30 | 82.80 | 70.90 | 45.00 | 29.80 |

scenarios conducted on the CIFAR10 and CIFAR100 datasets. Table 2 $\sim$ 11 display the performance outcomes on the training and test sets. In these experiments, we assessed the impact of assigning different scaled gradients to the model on mitigating catastrophic forgetting. The models used to conduct the experiments were all initialized with the same parameter, denoted as $\theta_0$. Coefficients $\alpha$ and $\beta$ denote the different proportions of the assigned gradients. $g_1$ represents the gradient of the well-trained model on the CIFAR10 dataset, and $g_2$ represents that on the CIFAR100 dataset.

Table 2 $\sim$ 11 can be divded into five distinct groups of experiments: (1) small-scale experiments (Table 2 and 3), (2) medium-scale experiments (Table 4 and 5), (3) large-scale experiments (Table 6 and 7), (4) large-scale experiments on the test set (Table 8 and 9), and (5) finegrained results of large-scale experiments on the test set (Table 10 and 11). The table within a group shows the results of the same model on CIFAR10 and CIFAR100, respectively.

Within each group, we sought to identify a pair of values for $\alpha$ and $\beta$ (highlighted in bold within the tables) that produced a model exhibiting excellent performance on both datasets. For instance, considering 10 and Table 11, when $\alpha$ is set to 0.6 and $\beta$ to 1.0, the model $\theta_0 + \alpha g_1 + \beta g_2$ achieves an accuracy exceeding 55 on both the CIFAR10 and CIFAR100 test sets, specifically 65.03 and 57.20, respectively. In comparison to the performance exhibited on the test sets in Table 5 (see the main text), our proposed method successfully accommodates both datasets.

### B.3.2 DETAILED RESULTS FOR MODEL FUSION

The following section presents the empirical results obtained from the model fusion experiments. These experiments encompass small, medium, and large-scale scenarios conducted on the CIFAR10 and CIFAR100 datasets. Table 12 $\sim$ 19 display the performance outcomes on the training and test sets. In these experiments, we explored the effectiveness of model fusion by combining well-trained models on the CIFAR10 and CIFAR100 datasets. The fusion process involved integrating the models using coefficients of weights fusion, denoted as $\alpha$ and $\beta$. All models shared a common initialization parameter, $\theta_0$. $\theta_1$ represents the well-trained model on the CIFAR10 dataset, and $\theta_2$ represents that on the CIFAR100 dataset.

Table 12 $\sim$ 19 can be divided into four groups of experiments: (1) small-scale experiments (Table 12 and 13), (2) medium-scale experiments (Table 14 and 15), (3) large-scale experiments (Table 16 and 17), (4) experiments on the test set (Table 18 and 19).

Similar to the findings from the gradients assignment experiments, the results of the model fusion experiments exhibit regular patterns. In these four groups of tables, the accuracies on CIFAR10 gradually increase towards the upper right corner of the table. Conversely, on CIFAR100, high accuracies are concentrated in the lower left corner of the table. This observation allows us to easily identify a set of $\alpha$ and $\beta$ values along the diagonal of the table, which effectively enable the fusion model to perform well on both datasets. Regarding the finegrained results of the model fusion experiments on the test set, we have provided a detailed description in the previous Section B.3.

Table 2: The results of putting the gradient-assigned model trained by **small-scale** on **training** set of **CIFAR10**.

| $\theta_0 + \alpha g_1 + \beta g_2$ | CIFAR10 | | | | | | | | | | |
|---|---|---|---|---|---|---|---|---|---|---|---|
| $\beta$ \\ $\alpha$ | 0.00 | 0.01 | 0.10 | 0.50 | 1.00 | 5.00 | 10.0 | 20.0 | 30.0 | 40.0 | 50.0 |
| 0.00 | 25.0 | 25.0 | 65.0 | 100.0 | 100.0 | 52.5 | 30.0 | 30.0 | 27.5 | 27.5 | 27.5 |
| 0.01 | 25.0 | 27.5 | 65.0 | 100.0 | 100.0 | 52.5 | 30.0 | 30.0 | 27.5 | 27.5 | 27.5 |
| 0.10 | 22.5 | 32.5 | 62.5 | 100.0 | 100.0 | 52.5 | 30.0 | 30.0 | 27.5 | 30.0 | 27.5 |
| 0.50 | 12.5 | 12.5 | 42.5 | **92.5** | 97.5 | 55.0 | 27.5 | 27.5 | 27.5 | 30.0 | 25.0 |
| 1.00 | 22.5 | 22.5 | 27.5 | 65.0 | 90.0 | 52.5 | 25.0 | 27.5 | 27.5 | 27.5 | 27.5 |
| 5.00 | 17.5 | 17.5 | 17.5 | 20.0 | 25.0 | 25.0 | 25.0 | 25.0 | 22.5 | 27.5 | 27.5 |
| 10.0 | 20.0 | 20.0 | 20.0 | 22.5 | 25.0 | 27.5 | 25.0 | 22.5 | 22.5 | 22.5 | 25.0 |
| 20.0 | 22.5 | 22.5 | 22.5 | 22.5 | 22.5 | 27.5 | 25.0 | 20.0 | 15.0 | 17.5 | 22.5 |
| 30.0 | 22.5 | 22.5 | 22.5 | 22.5 | 22.5 | 22.5 | 22.5 | 22.5 | 17.5 | 10.0 | 12.5 |
| 40.0 | 20.0 | 20.0 | 20.0 | 20.0 | 20.0 | 20.0 | 20.0 | 22.5 | 17.5 | 10.0 | 7.50 |
| 50.0 | 12.5 | 12.5 | 12.5 | 12.5 | 12.5 | 12.5 | 20.0 | 20.0 | 17.5 | 7.50 | 7.50 |

Table 3: The results of putting the gradient-assigned model trained by **small-scale** on **training** set of **CIFAR100**.

| $\theta_0 + \alpha g_1 + \beta g_2$ | CIFAR100 | | | | | | | | | | |
|---|---|---|---|---|---|---|---|---|---|---|---|
| $\beta$ \\ $\alpha$ | 0.00 | 0.01 | 0.10 | 0.50 | 1.00 | 5.00 | 10.0 | 20.0 | 30.0 | 40.0 | 50.0 |
| 0.00 | 17.5 | 17.5 | 15.0 | 15.0 | 15.0 | 17.5 | 10.0 | 17.5 | 20.0 | 17.5 | 15.0 |
| 0.01 | 17.5 | 17.5 | 17.5 | 20.0 | 15.0 | 17.5 | 10.0 | 17.5 | 20.0 | 17.5 | 15.0 |
| 0.10 | 42.5 | 42.5 | 40.0 | 37.5 | 15.0 | 17.5 | 10.0 | 17.5 | 20.0 | 17.5 | 15.0 |
| 0.50 | 100.0 | 100.0 | 100.0 | **85.0** | 52.5 | 20.0 | 12.5 | 17.5 | 20.0 | 17.5 | 15.0 |
| 1.00 | 100.0 | 100.0 | 100.0 | 100.0 | 72.5 | 17.5 | 12.5 | 15.0 | 20.0 | 17.5 | 15.0 |
| 5.00 | 62.5 | 62.5 | 62.5 | 65.0 | 60.0 | 25.0 | 25.0 | 15.0 | 17.5 | 17.5 | 15.0 |
| 10.0 | 50.0 | 50.0 | 47.5 | 47.5 | 47.5 | 32.5 | 17.5 | 22.5 | 17.5 | 17.5 | 15.0 |
| 20.0 | 35.0 | 35.0 | 35.0 | 37.5 | 35.0 | 30.0 | 22.5 | 17.5 | 20.0 | 22.5 | 10.0 |
| 30.0 | 25.0 | 25.0 | 25.0 | 25.0 | 25.0 | 27.5 | 22.5 | 17.5 | 17.5 | 22.5 | 20.0 |
| 40.0 | 22.5 | 22.5 | 22.5 | 22.5 | 22.5 | 20.0 | 20.0 | 25.0 | 17.5 | 15.0 | 20.0 |
| 50.0 | 22.5 | 22.5 | 22.5 | 22.5 | 22.5 | 20.0 | 15.0 | 22.5 | 15.0 | 15.0 | 12.5 |

These findings demonstrate the efficacy of our approach in handling catastrophic forgetting. By appropriately fusing the models using suitable values of $\alpha$ and $\beta$, we attain a model that performs exceptionally well on both CIFAR10 and CIFAR100. Furthermore, when comparing these results to those obtained using the traditional approach outlined in Table 5 (see the main text), it becomes evident that our proposed method effectively overcomes the limitations of catastrophic forgetting, thereby accommodating the unique characteristics of both datasets.

Table 4: The results of putting the gradient-assigned model trained by **medium-scale** on **training** set of **CIFAR10**.

| $\theta_0 + \alpha g_1 + \beta g_2$ | CIFAR10 | | | | | | | | | | |
|---|---|---|---|---|---|---|---|---|---|---|---|
| $\beta$ \ $\alpha$ | 0.00 | 0.01 | 0.10 | 0.50 | 1.00 | 5.00 | 10.0 | 20.0 | 30.0 | 40.0 | 50.0 |
| 0.00 | 10.6 | 11.7 | 26.9 | 97.5 | 100.0 | 55.6 | 25.5 | 15.5 | 14.2 | 14.0 | 13.1 |
| 0.01 | 11.2 | 12.2 | 27.0 | 97.5 | 100.0 | 55.5 | 25.5 | 15.5 | 14.2 | 14.0 | 13.1 |
| 0.10 | 12.0 | 13.0 | 24.5 | 96.6 | 100.0 | 54.7 | 24.5 | 15.2 | 13.9 | 13.7 | 13.1 |
| 0.50 | 14.4 | 14.5 | 15.6 | **44.7** | 98.5 | 50.3 | 23.3 | 14.2 | 13.5 | 13.2 | 13.0 |
| 1.00 | 13.9 | 14.0 | 14.5 | 17.6 | 43.2 | 44.2 | 21.2 | 12.9 | 13.1 | 12.6 | 12.8 |
| 5.00 | 12.3 | 12.3 | 12.1 | 12.2 | 13.0 | 14.1 | 12.4 | 11.2 | 10.4 | 11.3 | 12.0 |
| 10.0 | 12.0 | 12.0 | 12.3 | 12.8 | 12.5 | 11.8 | 11.8 | 10.6 | 11.6 | 11.4 | 10.6 |
| 20.0 | 11.3 | 11.2 | 11.0 | 11.3 | 11.3 | 11.6 | 11.2 | 10.6 | 11.1 | 12.0 | 12.4 |
| 30.0 | 11.0 | 11.0 | 11.1 | 11.2 | 11.4 | 11.7 | 10.5 | 10.8 | 10.7 | 11.9 | 12.9 |
| 40.0 | 10.8 | 10.8 | 10.8 | 10.9 | 11.1 | 10.6 | 11.2 | 10.8 | 11.1 | 11.9 | 12.1 |
| 50.0 | 10.2 | 10.2 | 10.2 | 10.3 | 10.2 | 10.6 | 10.4 | 11.0 | 11.4 | 12.2 | 12.9 |

Table 5: The results of putting the gradient-assigned model trained by **medium-scale** on **training** set of **CIFAR100**.

| $\theta_0 + \alpha g_1 + \beta g_2$ | CIFAR100 | | | | | | | | | | |
|---|---|---|---|---|---|---|---|---|---|---|---|
| $\beta$ \ $\alpha$ | 0.00 | 0.01 | 0.10 | 0.50 | 1.00 | 5.00 | 10.0 | 20.0 | 30.0 | 40.0 | 50.0 |
| 0.00 | 15.6 | 16.1 | 15.5 | 9.5 | 9.1 | 5.3 | 4.1 | 4.7 | 4.8 | 4.9 | 5.9 |
| 0.01 | 16.5 | 17.1 | 17.9 | 10.1 | 9.2 | 5.3 | 4.1 | 4.7 | 4.8 | 4.8 | 5.9 |
| 0.10 | 40.7 | 41.3 | 39.2 | 13.6 | 10.0 | 5.2 | 4.2 | 4.7 | 4.7 | 4.7 | 5.8 |
| 0.50 | 98.2 | 98.2 | 98.3 | **81.8** | 23.3 | 5.5 | 4.0 | 4.7 | 4.7 | 4.8 | 6.2 |
| 1.00 | 100.0 | 100.0 | 100.0 | 99.7 | 81.8 | 6.4 | 4.2 | 4.6 | 4.7 | 4.9 | 6.1 |
| 5.00 | 57.5 | 57.6 | 57.5 | 54.8 | 50.9 | 21.1 | 9.1 | 5.8 | 5.8 | 5.9 | 5.9 |
| 10.0 | 33.3 | 33.2 | 32.9 | 31.7 | 30.9 | 21.7 | 14.6 | 9.7 | 7.0 | 7.4 | 7.0 |
| 20.0 | 22.0 | 22.0 | 22.2 | 21.8 | 22.1 | 20.8 | 16.4 | 11.4 | 10.6 | 9.5 | 9.5 |
| 30.0 | 21.2 | 21.2 | 21.2 | 21.3 | 21.5 | 19.7 | 18.2 | 12.9 | 12.5 | 11.4 | 10.3 |
| 40.0 | 21.6 | 21.6 | 21.6 | 21.4 | 21.1 | 20.6 | 19.4 | 15.3 | 13.0 | 13.6 | 11.8 |
| 50.0 | 20.9 | 20.9 | 20.9 | 21.2 | 21.2 | 20.4 | 19.0 | 16.6 | 13.9 | 13.1 | 12.5 |

Table 6: The results of putting the gradient-assigned model trained by **large-scale** on **training** set of **CIFAR10**.

| $\theta_0 + \alpha g_1 + \beta g_2$ | CIFAR10 | | | | | | | | | | |
|---|---|---|---|---|---|---|---|---|---|---|---|
| $\beta$ \ $\alpha$ | 0.00 | 0.01 | 0.10 | 0.50 | 1.00 | 5.00 | 10.00 | 20.00 | 30.00 | 40.00 | 50.00 |
| 0.00 | 12.51 | 13.34 | 31.83 | 97.67 | 100.00 | 44.83 | 12.92 | 10.63 | 10.28 | 10.24 | 10.20 |
| 0.01 | 12.54 | 13.34 | 31.84 | 97.71 | 100.00 | 44.76 | 12.91 | 10.63 | 10.28 | 10.25 | 10.20 |
| 0.10 | 13.30 | 14.09 | 30.50 | 97.95 | 100.00 | 44.28 | 12.89 | 10.63 | 10.28 | 10.25 | 10.20 |
| 0.50 | 11.95 | 12.09 | 14.51 | 93.80 | 99.86 | 42.09 | 12.79 | 10.62 | 10.26 | 10.23 | 10.20 |
| 1.00 | 10.95 | 11.03 | 11.83 | **49.10** | 94.42 | 39.25 | 12.66 | 10.63 | 10.26 | 10.24 | 10.19 |
| 5.00 | 9.49 | 9.49 | 9.50 | 9.49 | 9.42 | 19.10 | 11.70 | 10.60 | 10.24 | 10.20 | 10.13 |
| 10.0 | 9.06 | 9.05 | 8.98 | 8.84 | 8.43 | 10.23 | 10.72 | 10.51 | 10.19 | 10.16 | 10.12 |
| 20.0 | 9.31 | 9.31 | 9.24 | 8.94 | 8.67 | 10.00 | 9.64 | 10.32 | 10.14 | 10.06 | 10.01 |
| 30.0 | 9.44 | 9.43 | 9.48 | 9.80 | 10.16 | 9.99 | 9.38 | 10.03 | 9.99 | 9.95 | 9.89 |
| 40.0 | 10.29 | 10.30 | 10.33 | 10.70 | 10.93 | 9.94 | 9.53 | 9.60 | 9.88 | 9.79 | 9.80 |
| 50.0 | 10.30 | 10.31 | 10.36 | 10.47 | 10.63 | 9.82 | 9.40 | 9.24 | 9.56 | 9.66 | 9.64 |

Table 7: The results of putting the gradient-assigned model trained by **large-scale** on **training** set of **CIFAR100**.

| $\theta_0 + \alpha g_1 + \beta g_2$ | CIFAR100 | | | | | | | | | | |
|---|---|---|---|---|---|---|---|---|---|---|---|
| $\beta$ \ $\alpha$ | 0.00 | 0.01 | 0.10 | 0.50 | 1.00 | 5.00 | 10.00 | 20.00 | 30.00 | 40.00 | 50.00 |
| 0.00 | 14.68 | 14.18 | 12.08 | 9.44 | 7.02 | 9.34 | 14.74 | 9.32 | 9.84 | 11.02 | 11.68 |
| 0.01 | 16.46 | 16.16 | 13.52 | 9.54 | 7.14 | 9.36 | 14.74 | 9.32 | 9.84 | 11.02 | 11.68 |
| 0.10 | 39.70 | 39.96 | 34.32 | 11.02 | 7.56 | 9.58 | 14.66 | 9.28 | 9.82 | 11.00 | 11.66 |
| 0.50 | 98.18 | 98.28 | 97.28 | 31.48 | 11.36 | 10.40 | 14.64 | 9.32 | 9.84 | 10.94 | 11.62 |
| 1.00 | 100.00 | 100.00 | 99.98 | **77.82** | 23.70 | 11.42 | 14.60 | 9.40 | 9.86 | 11.00 | 11.60 |
| 5.00 | 68.04 | 68.22 | 68.86 | 69.22 | 66.52 | 16.94 | 13.34 | 9.62 | 9.90 | 11.40 | 11.52 |
| 10.0 | 49.72 | 49.86 | 50.28 | 52.48 | 52.64 | 23.40 | 12.44 | 9.60 | 10.30 | 11.12 | 11.44 |
| 20.0 | 37.06 | 37.04 | 37.00 | 36.88 | 35.80 | 20.00 | 12.14 | 9.98 | 10.18 | 10.54 | 11.38 |
| 30.0 | 23.98 | 23.92 | 23.80 | 22.90 | 21.74 | 18.84 | 11.26 | 10.56 | 10.18 | 10.20 | 10.80 |
| 40.0 | 19.30 | 19.22 | 19.28 | 19.28 | 19.26 | 18.18 | 11.24 | 11.14 | 10.58 | 9.96 | 10.18 |
| 50.0 | 18.70 | 18.72 | 18.66 | 18.62 | 18.48 | 16.44 | 11.22 | 11.02 | 11.64 | 10.16 | 10.04 |

Table 8: The results of putting the gradient-assigned model trained by **large-scale** on **testing** set of **CIFAR10**.

| $\theta_0 + \alpha g_1 + \beta g_2$ | CIFAR10-test | | | | | | | | | | |
|---|---|---|---|---|---|---|---|---|---|---|---|
| $\beta$ \ $\alpha$ | 0.00 | 0.01 | 0.10 | 0.50 | 1.00 | 5.00 | 10.00 | 20.00 | 30.00 | 40.00 | 50.00 |
| 0.00 | 12.04 | 12.66 | 26.77 | 83.02 | 87.33 | 56.89 | 15.41 | 10.16 | 9.46 | 8.97 | 8.98 |
| 0.01 | 11.99 | 12.59 | 26.81 | 83.03 | 87.34 | 56.86 | 15.40 | 10.16 | 9.46 | 8.97 | 8.98 |
| 0.10 | 12.18 | 12.94 | 25.19 | 83.07 | 87.32 | 56.56 | 15.40 | 10.16 | 9.45 | 8.96 | 8.98 |
| 0.50 | 10.71 | 10.96 | 13.41 | 79.74 | 86.14 | 55.43 | 15.39 | 10.10 | 9.45 | 8.98 | 9.00 |
| 1.00 | 10.45 | 10.49 | 11.13 | **45.54** | 81.01 | 53.87 | 15.28 | 10.10 | 9.42 | 8.98 | 9.01 |
| 5.00 | 11.52 | 11.58 | 11.65 | 12.55 | 13.96 | 39.16 | 14.47 | 10.09 | 9.32 | 9.01 | 9.08 |
| 10.0 | 11.86 | 11.81 | 11.91 | 11.94 | 12.34 | 17.22 | 13.23 | 9.93 | 9.05 | 9.03 | 8.87 |
| 20.0 | 11.88 | 11.82 | 11.70 | 11.53 | 11.33 | 10.39 | 10.58 | 9.53 | 9.13 | 8.83 | 8.66 |
| 30.0 | 10.92 | 10.89 | 10.90 | 10.71 | 10.60 | 10.08 | 10.11 | 9.42 | 8.71 | 8.61 | 8.54 |
| 40.0 | 9.17 | 9.19 | 9.33 | 9.23 | 9.32 | 9.11 | 9.33 | 9.29 | 8.61 | 8.28 | 8.59 |
| 50.0 | 8.50 | 8.50 | 8.51 | 8.54 | 8.51 | 8.39 | 8.50 | 8.86 | 8.48 | 8.25 | 8.38 |

Table 9: The results of putting the gradient-assigned model trained by **large-scale** on **testing** set of **CIFAR100**.

| $\theta_0 + \alpha g_1 + \beta g_2$ | CIFAR100-test | | | | | | | | | | |
|---|---|---|---|---|---|---|---|---|---|---|---|
| $\beta$ \ $\alpha$ | 0.00 | 0.01 | 0.10 | 0.50 | 1.00 | 5.00 | 10.00 | 20.00 | 30.00 | 40.00 | 50.00 |
| 0.00 | 15.00 | 14.60 | 11.50 | 11.60 | 9.10 | 6.60 | 10.90 | 12.30 | 10.90 | 9.80 | 9.30 |
| 0.01 | 16.90 | 16.80 | 12.70 | 11.80 | 9.10 | 6.60 | 10.90 | 12.30 | 10.90 | 9.80 | 9.30 |
| 0.10 | 40.30 | 40.20 | 32.20 | 13.60 | 9.60 | 6.60 | 10.90 | 12.30 | 10.90 | 9.80 | 9.30 |
| 0.50 | 81.50 | 82.00 | 81.60 | 29.90 | 12.60 | 6.10 | 11.00 | 12.20 | 10.90 | 9.80 | 9.30 |
| 1.00 | 87.00 | 86.90 | 87.50 | **72.70** | 20.00 | 6.60 | 11.00 | 12.30 | 10.70 | 9.80 | 9.30 |
| 5.00 | 68.20 | 68.10 | 68.00 | 67.10 | 65.30 | 10.40 | 11.60 | 12.00 | 10.80 | 10.50 | 10.00 |
| 10.0 | 58.20 | 58.10 | 58.20 | 58.80 | 59.20 | 19.30 | 12.60 | 11.30 | 11.10 | 10.50 | 10.10 |
| 20.0 | 48.30 | 48.30 | 48.10 | 46.80 | 45.40 | 22.80 | 13.40 | 10.50 | 11.20 | 10.00 | 9.50 |
| 30.0 | 36.10 | 36.10 | 36.00 | 35.40 | 32.10 | 22.30 | 14.50 | 10.20 | 9.50 | 10.40 | 9.30 |
| 40.0 | 26.70 | 26.60 | 26.40 | 24.70 | 24.30 | 21.50 | 16.10 | 9.60 | 9.60 | 9.30 | 9.50 |
| 50.0 | 22.90 | 22.90 | 22.90 | 23.20 | 22.30 | 19.10 | 15.70 | 9.60 | 10.00 | 10.00 | 9.60 |

Table 10: The **finegrained** results of putting the gradient-assigned model trained by **large-scale** on **testing** set of **CIFAR10**.

| $\theta_0 + \alpha g_1 + \beta g_2$ | CIFAR10-test-finegrained | | | | | | | | | |
|---|---|---|---|---|---|---|---|---|---|---|
| $\beta$ \ $\alpha$ | 0.50 | 0.60 | 0.70 | 0.80 | 0.90 | 1.00 | 2.00 | 3.00 | 4.00 | 5.00 |
| 0.50 | 79.74 | 83.60 | 85.20 | 86.09 | 86.01 | 86.14 | 82.25 | 77.29 | 69.17 | 55.43 |
| 0.60 | 76.89 | 82.08 | 84.20 | 85.25 | 85.43 | 85.43 | 81.83 | 77.03 | 68.94 | 55.24 |
| 0.70 | 72.16 | 79.79 | 82.84 | 84.16 | 84.47 | 84.61 | 81.44 | 76.79 | 68.55 | 54.95 |
| 0.80 | 65.38 | 76.58 | 80.84 | 82.66 | 83.46 | 83.75 | 81.04 | 76.58 | 68.24 | 54.64 |
| 0.90 | 56.05 | 71.57 | 78.34 | 80.85 | 81.97 | 82.65 | 80.66 | 76.26 | 67.97 | 54.31 |
| 1.00 | 45.54 | **65.03** | 74.42 | 78.26 | 79.98 | 81.01 | 80.29 | 76.00 | 67.79 | 53.87 |
| 2.00 | 15.28 | 17.20 | 20.67 | 26.97 | 35.59 | 46.09 | 74.07 | 72.09 | 64.19 | 50.64 |
| 3.00 | 13.20 | 13.73 | 14.46 | 15.57 | 16.91 | 19.07 | 61.90 | 66.98 | 60.11 | 47.44 |
| 4.00 | 13.03 | 13.17 | 13.38 | 13.75 | 14.31 | 14.96 | 41.77 | 59.30 | 54.75 | 43.25 |
| 5.00 | 12.55 | 12.78 | 12.94 | 13.20 | 13.65 | 13.96 | 26.18 | 47.60 | 48.35 | 39.16 |

Table 11: The **finegrained** results of putting the gradient-assigned model trained by **large-scale** on **testing** set of **CIFAR100**.

| $\theta_0 + \alpha g_1 + \beta g_2$ | CIFAR100-test-finegrained | | | | | | | | | |
|---|---|---|---|---|---|---|---|---|---|---|
| $\beta$ \ $\alpha$ | 0.50 | 0.60 | 0.70 | 0.80 | 0.90 | 1.00 | 2.00 | 3.00 | 4.00 | 5.00 |
| 0.50 | 29.90 | 22.50 | 18.00 | 14.90 | 13.80 | 12.60 | 8.70 | 7.70 | 7.20 | 6.10 |
| 0.60 | 38.80 | 27.30 | 21.00 | 17.20 | 15.10 | 13.60 | 9.10 | 7.50 | 7.30 | 6.30 |
| 0.70 | 51.10 | 33.80 | 24.30 | 20.10 | 16.90 | 15.00 | 9.30 | 7.70 | 7.30 | 6.20 |
| 0.80 | 61.40 | 41.60 | 30.20 | 22.90 | 18.40 | 16.60 | 9.50 | 7.90 | 7.40 | 6.40 |
| 0.90 | 68.30 | 50.60 | 35.20 | 26.90 | 20.60 | 18.50 | 9.80 | 8.30 | 7.50 | 6.50 |
| 1.00 | 72.70 | **57.20** | 42.50 | 31.60 | 24.20 | 20.00 | 9.80 | 8.60 | 7.40 | 6.60 |
| 2.00 | 75.80 | 73.40 | 69.60 | 63.10 | 58.80 | 53.20 | 14.00 | 9.80 | 8.00 | 7.20 |
| 3.00 | 73.40 | 72.10 | 69.40 | 67.80 | 65.60 | 64.00 | 26.00 | 12.20 | 9.50 | 8.20 |
| 4.00 | 69.90 | 69.20 | 68.50 | 67.10 | 66.00 | 65.10 | 41.50 | 17.20 | 11.20 | 9.50 |
| 5.00 | 67.10 | 66.70 | 66.50 | 66.10 | 65.20 | 65.30 | 51.10 | 24.80 | 13.80 | 10.40 |

Table 12: The results of putting the fusion model trained by **small-scale** on **training** set of **CIFAR10**.

| $\alpha\theta_1 + \beta\theta_2$ | CIFAR10 | | | | | | | | | |
|---|---|---|---|---|---|---|---|---|---|---|
| $\beta$ \ $\alpha$ | 0.00 | 0.01 | 0.10 | 0.50 | 1.00 | 5.00 | 10.0 | 20.0 | 30.0 | 40.0 | 50.0 |
| 0.00 | 10.0 | 10.0 | 100.0 | 100.0 | 100.0 | 100.0 | 100.0 | 100.0 | 100.0 | 100.0 | 100.0 |
| 0.01 | 10.0 | 10.0 | 100.0 | 100.0 | 100.0 | 100.0 | 100.0 | 100.0 | 100.0 | 100.0 | 100.0 |
| 0.10 | 15.0 | 20.0 | 92.5 | 100.0 | 100.0 | 100.0 | 100.0 | 100.0 | 100.0 | 100.0 | 100.0 |
| 0.50 | 20.0 | 22.5 | 30.0 | **92.5** | 97.5 | 100.0 | 100.0 | 100.0 | 100.0 | 100.0 | 100.0 |
| 1.00 | 22.5 | 22.5 | 27.5 | 85.0 | 92.5 | 100.0 | 100.0 | 100.0 | 100.0 | 100.0 | 100.0 |
| 5.00 | 22.5 | 22.5 | 22.5 | 27.5 | 30.0 | 92.5 | 97.5 | 100.0 | 100.0 | 100.0 | 100.0 |
| 10.0 | 22.5 | 22.5 | 22.5 | 25.0 | 27.5 | 85.0 | 92.5 | 97.5 | 100.0 | 100.0 | 100.0 |
| 20.0 | 22.5 | 22.5 | 22.5 | 25.0 | 25.0 | 40.0 | 85.0 | 92.5 | 95.0 | 97.5 | 97.5 |
| 30.0 | 22.5 | 22.5 | 22.5 | 22.5 | 25.0 | 30.0 | 60.0 | 85.0 | 92.5 | 95.0 | 95.0 |
| 40.0 | 22.5 | 22.5 | 22.5 | 22.5 | 25.0 | 27.5 | 40.0 | 85.0 | 87.5 | 92.5 | 95.0 |
| 50.0 | 22.5 | 22.5 | 22.5 | 22.5 | 22.5 | 27.5 | 30.0 | 67.5 | 85.0 | 90.0 | 92.5 |

Table 13: The results of putting the fusion model trained by **small-scale** on **training** set of **CIFAR100**.

| $\alpha\theta_1 + \beta\theta_2$ | CIFAR100 | | | | | | | | | | |
|---|---|---|---|---|---|---|---|---|---|---|---|
| $\beta$ \ $\alpha$ | 0.00 | 0.01 | 0.10 | 0.50 | 1.00 | 5.00 | 10.0 | 20.0 | 30.0 | 40.0 | 50.0 |
| 0.00 | 10.0 | 10.0 | 15.0 | 15.0 | 15.0 | 15.0 | 15.0 | 15.0 | 15.0 | 15.0 | 15.0 |
| 0.01 | 10.0 | 10.0 | 22.5 | 15.0 | 15.0 | 15.0 | 15.0 | 15.0 | 15.0 | 15.0 | 15.0 |
| 0.10 | 100.0 | 100.0 | 85.0 | 22.5 | 15.0 | 15.0 | 15.0 | 15.0 | 15.0 | 15.0 | 15.0 |
| 0.50 | 100.0 | 100.0 | 100.0 | **85.0** | 67.5 | 15.0 | 15.0 | 15.0 | 15.0 | 15.0 | 15.0 |
| 1.00 | 100.0 | 100.0 | 100.0 | 100.0 | 82.5 | 25.0 | 15.0 | 15.0 | 15.0 | 15.0 | 15.0 |
| 5.00 | 100.0 | 100.0 | 100.0 | 100.0 | 100.0 | 82.5 | 65.0 | 30.0 | 20.0 | 15.0 | 15.0 |
| 10.0 | 100.0 | 100.0 | 100.0 | 100.0 | 100.0 | 100.0 | 82.5 | 65.0 | 47.5 | 30.0 | 25.0 |
| 20.0 | 100.0 | 100.0 | 100.0 | 100.0 | 100.0 | 100.0 | 100.0 | 82.5 | 75.0 | 65.0 | 50.0 |
| 30.0 | 100.0 | 100.0 | 100.0 | 100.0 | 100.0 | 100.0 | 100.0 | 100.0 | 82.5 | 77.5 | 70.0 |
| 40.0 | 100.0 | 100.0 | 100.0 | 100.0 | 100.0 | 100.0 | 100.0 | 100.0 | 97.5 | 82.5 | 80.0 |
| 50.0 | 100.0 | 100.0 | 100.0 | 100.0 | 100.0 | 100.0 | 100.0 | 100.0 | 100.0 | 95.0 | 82.5 |

Table 14: The results of putting the fusion model trained by **medium-scale** on **training** set of **CIFAR10**.

| $\alpha\theta_1 + \beta\theta_2$ | CIFAR10 | | | | | | | | | | |
|---|---|---|---|---|---|---|---|---|---|---|---|
| $\beta$ \ $\alpha$ | 0.00 | 0.01 | 0.10 | 0.50 | 1.00 | 5.00 | 10.0 | 20.0 | 30.0 | 40.0 | 50.0 |
| 0.00 | 10.0 | 10.0 | 100.0 | 100.0 | 100.0 | 100.0 | 100.0 | 100.0 | 100.0 | 100.0 | 100.0 |
| 0.01 | 10.0 | 10.0 | 100.0 | 100.0 | 100.0 | 100.0 | 100.0 | 100.0 | 100.0 | 100.0 | 100.0 |
| 0.10 | 14.7 | 14.9 | **45.2** | 100.0 | 100.0 | 100.0 | 100.0 | 100.0 | 100.0 | 100.0 | 100.0 |
| 0.50 | 13.8 | 14.1 | 14.8 | 44.7 | 96.6 | 100.0 | 100.0 | 100.0 | 100.0 | 100.0 | 100.0 |
| 1.00 | 13.9 | 13.9 | 14.6 | 18.6 | 44.5 | 100.0 | 100.0 | 100.0 | 100.0 | 100.0 | 100.0 |
| 5.00 | 13.8 | 13.8 | 13.9 | 14.5 | 14.9 | 44.7 | 96.6 | 100.0 | 100.0 | 100.0 | 100.0 |
| 10.0 | 13.8 | 13.7 | 13.8 | 14.0 | 14.5 | 18.4 | 44.7 | 96.6 | 99.7 | 100.0 | 100.0 |
| 20.0 | 13.8 | 13.7 | 13.8 | 14.0 | 14.0 | 14.8 | 18.4 | 44.7 | 84.6 | 96.6 | 99.3 |
| 30.0 | 13.7 | 13.8 | 13.8 | 13.8 | 13.9 | 14.8 | 15.7 | 22.4 | 44.7 | 75.6 | 90.5 |
| 40.0 | 13.7 | 13.8 | 13.8 | 13.8 | 14.0 | 14.7 | 14.8 | 18.4 | 26.3 | 44.7 | 68.9 |
| 50.0 | 13.8 | 13.8 | 13.8 | 13.8 | 13.8 | 14.5 | 14.7 | 16.6 | 20.7 | 28.4 | 44.7 |

Table 15: The results of putting the fusion model trained by **medium-scale** on **training** set of **CIFAR100**.

| $\alpha\theta_1 + \beta\theta_2$ | CIFAR100 | | | | | | | | | | |
|---|---|---|---|---|---|---|---|---|---|---|---|
| $\beta$ \ $\alpha$ | 0.00 | 0.01 | 0.10 | 0.50 | 1.00 | 5.00 | 10.0 | 20.0 | 30.0 | 40.0 | 50.0 |
| 0.00 | 10.0 | 10.0 | 8.9 | 9.0 | 9.1 | 9.1 | 9.2 | 9.2 | 9.2 | 9.2 | 9.2 |
| 0.01 | 10.0 | 10.0 | 10.5 | 9.1 | 9.1 | 9.1 | 9.2 | 9.2 | 9.2 | 9.2 | 9.2 |
| 0.10 | 100.0 | 100.0 | **82.6** | 12.8 | 10.4 | 9.2 | 9.2 | 9.2 | 9.2 | 9.2 | 9.2 |
| 0.50 | 100.0 | 100.0 | 100.0 | 81.8 | 26.2 | 10.3 | 9.5 | 9.2 | 9.2 | 9.2 | 9.2 |
| 1.00 | 100.0 | 100.0 | 100.0 | 99.2 | 81.8 | 13.2 | 10.3 | 9.5 | 9.1 | 9.2 | 9.2 |
| 5.00 | 100.0 | 100.0 | 100.0 | 100.0 | 100.0 | 81.7 | 26.4 | 14.3 | 11.8 | 10.7 | 10.3 |
| 10.0 | 100.0 | 100.0 | 100.0 | 100.0 | 100.0 | 99.2 | 81.6 | 26.4 | 17.4 | 14.3 | 13.2 |
| 20.0 | 100.0 | 100.0 | 100.0 | 100.0 | 100.0 | 99.8 | 99.2 | 81.6 | 44.8 | 26.4 | 19.4 |
| 30.0 | 100.0 | 100.0 | 100.0 | 100.0 | 100.0 | 100.0 | 99.8 | 96.1 | 81.6 | 55.9 | 36.7 |
| 40.0 | 100.0 | 100.0 | 100.0 | 100.0 | 100.0 | 100.0 | 99.8 | 99.2 | 94.1 | 81.6 | 63.1 |
| 50.0 | 100.0 | 100.0 | 100.0 | 100.0 | 100.0 | 100.0 | 100.0 | 99.6 | 98.1 | 92.4 | 81.6 |

Table 16: The results of putting the fusion model trained by **large-scale** on **training** set of **CIFAR10**.

| $\alpha\theta_1 + \beta\theta_2$ | CIFAR10 | | | | | | | | | | |
|---|---|---|---|---|---|---|---|---|---|---|---|
| $\beta$ \ $\alpha$ | 0.00 | 0.01 | 0.10 | 0.50 | 1.00 | 5.00 | 10.00 | 20.00 | 30.00 | 40.00 | 50.00 |
| 0.00 | 10.00 | 10.00 | 100.00 | 100.00 | 100.00 | 100.00 | 100.00 | 100.00 | 100.00 | 100.00 | 100.00 |
| 0.01 | 10.00 | 10.00 | 100.00 | 100.00 | 100.00 | 100.00 | 100.00 | 100.00 | 100.00 | 100.00 | 100.00 |
| 0.10 | 11.32 | 12.18 | 93.61 | 99.99 | 100.00 | 100.00 | 100.00 | 100.00 | 100.00 | 100.00 | 100.00 |
| 0.50 | 10.97 | 11.10 | 13.37 | 93.80 | 99.85 | 100.00 | 100.00 | 100.00 | 100.00 | 100.00 | 100.00 |
| 1.00 | 10.95 | 11.03 | 11.96 | 44.27 | 93.84 | 99.99 | 100.00 | 100.00 | 100.00 | 100.00 | 100.00 |
| 5.00 | 10.93 | 10.95 | 11.10 | 11.92 | 13.36 | 93.85 | 99.86 | 99.99 | 100.00 | 100.00 | 100.00 |
| 10.0 | 10.93 | 10.94 | 11.00 | 11.36 | 11.91 | 44.31 | 93.84 | 99.86 | 99.97 | 99.99 | 99.99 |
| 20.0 | 10.93 | 10.94 | 10.94 | 11.15 | 11.36 | 14.76 | 44.31 | 93.84 | 99.17 | 99.86 | 99.95 |
| 30.0 | 10.94 | 10.93 | 10.95 | 11.05 | 11.23 | 12.85 | 19.07 | **73.19** | 93.84 | 98.45 | 99.55 |
| 40.0 | 10.94 | 10.93 | 10.94 | 11.01 | 11.15 | 12.28 | 14.75 | 44.31 | 81.54 | 93.84 | 97.81 |
| 50.0 | 10.94 | 10.93 | 10.94 | 11.01 | 11.09 | 11.91 | 13.38 | 26.25 | 63.55 | 85.13 | 93.84 |

Table 17: The results of putting the fusion model trained by **large-scale** on **training** set of **CIFAR100**.

| $\alpha\theta_1 + \beta\theta_2$ | CIFAR100 | | | | | | | | | | |
|---|---|---|---|---|---|---|---|---|---|---|---|
| $\beta$ \ $\alpha$ | 0.00 | 0.01 | 0.10 | 0.50 | 1.00 | 5.00 | 10.00 | 20.00 | 30.00 | 40.00 | 50.00 |
| 0.00 | 10.00 | 10.00 | 7.82 | 7.14 | 7.02 | 7.06 | 7.00 | 7.00 | 7.00 | 7.00 | 7.00 |
| 0.01 | 10.00 | 10.00 | 8.46 | 7.18 | 7.04 | 7.04 | 7.04 | 7.00 | 7.00 | 7.00 | 7.00 |
| 0.10 | 99.98 | 99.98 | 31.80 | 8.44 | 7.56 | 7.04 | 7.04 | 7.02 | 7.04 | 7.08 | 7.04 |
| 0.50 | 100.00 | 100.00 | 99.86 | 31.48 | 12.16 | 7.56 | 7.22 | 7.08 | 7.04 | 7.02 | 7.00 |
| 1.00 | 100.00 | 100.00 | 100.00 | 82.76 | 31.26 | 8.30 | 7.56 | 7.20 | 7.12 | 7.08 | 7.04 |
| 5.00 | 100.00 | 100.00 | 100.00 | 100.00 | 99.86 | 31.30 | 12.08 | 8.60 | 8.10 | 7.76 | 7.58 |
| 10.0 | 100.00 | 100.00 | 100.00 | 100.00 | 100.00 | 82.70 | 31.30 | 12.06 | 9.76 | 8.60 | 8.30 |
| 20.0 | 100.00 | 100.00 | 100.00 | 100.00 | 100.00 | 99.54 | 82.68 | 31.30 | 16.44 | 12.06 | 10.48 |
| 30.0 | 100.00 | 100.00 | 100.00 | 100.00 | 100.00 | 99.92 | 97.56 | **60.62** | 31.30 | 19.14 | 14.70 |
| 40.0 | 100.00 | 100.00 | 100.00 | 100.00 | 100.00 | 99.96 | 99.54 | 82.68 | 50.82 | 31.30 | 21.26 |
| 50.0 | 100.00 | 100.00 | 100.00 | 100.00 | 100.00 | 100.00 | 99.86 | 93.62 | 69.24 | 46.38 | 31.30 |

Table 18: The results of putting the fusion model trained by **large-scale** on **testing** set of **CIFAR10**.

| $\alpha\theta_1 + \beta\theta_2$ | CIFAR10-test | | | | | | | | | | |
|---|---|---|---|---|---|---|---|---|---|---|---|
| $\beta$ \ $\alpha$ | 0.00 | 0.01 | 0.10 | 0.50 | 1.00 | 5.00 | 10.00 | 20.00 | 30.00 | 40.00 | 50.00 |
| 0.00 | 10.00 | 10.00 | 87.29 | 87.28 | 87.33 | 87.33 | 87.31 | 87.32 | 87.32 | 87.32 | 87.32 |
| 0.01 | 10.00 | 10.00 | 87.22 | 87.18 | 87.29 | 87.32 | 87.32 | 87.32 | 87.32 | 87.33 | 87.33 |
| 0.10 | 9.60 | 10.58 | 79.44 | 86.99 | 87.25 | 87.35 | 87.34 | 87.33 | 87.33 | 87.33 | 87.33 |
| 0.50 | 10.36 | 10.46 | 12.85 | 79.74 | 85.80 | 87.17 | 87.29 | 87.33 | 87.33 | 87.32 | 87.34 |
| 1.00 | 10.45 | 10.49 | 11.17 | 37.99 | 79.77 | 87.09 | 87.17 | 87.29 | 87.33 | 87.33 | 87.35 |
| 5.00 | 10.51 | 10.51 | 10.54 | 11.23 | 12.87 | 79.77 | 85.86 | 86.97 | 87.15 | 87.22 | 87.17 |
| 10.0 | 10.50 | 10.53 | 10.53 | 10.80 | 11.24 | 38.06 | 79.77 | 85.87 | 86.76 | 86.97 | 87.09 |
| 20.0 | 10.51 | 10.52 | 10.55 | 10.60 | 10.78 | 14.37 | 38.06 | 79.78 | 84.54 | 85.87 | 86.41 |
| 30.0 | 10.52 | 10.52 | 10.55 | 10.54 | 10.70 | 12.08 | 18.74 | **63.11** | 79.80 | 83.63 | 85.13 |
| 40.0 | 10.52 | 10.52 | 10.55 | 10.55 | 10.60 | 11.45 | 14.38 | 38.06 | 70.51 | 79.81 | 82.94 |
| 50.0 | 10.52 | 10.52 | 10.55 | 10.54 | 10.54 | 11.24 | 12.87 | 24.04 | 54.48 | 73.42 | 79.81 |

Table 19: The results of putting the fusion model trained by **large-scale** on **testing** set of **CIFAR100**.

| $\alpha\theta_1 + \beta\theta_2$ | CIFAR100-test | | | | | | | | | | |
|---|---|---|---|---|---|---|---|---|---|---|---|
| $\beta$ \ $\alpha$ | 0.00 | 0.01 | 0.10 | 0.50 | 1.00 | 5.00 | 10.00 | 20.00 | 30.00 | 40.00 | 50.00 |
| 0.00 | 10.00 | 10.00 | 9.00 | 9.00 | 9.10 | 9.30 | 9.30 | 9.30 | 9.30 | 9.30 | 9.30 |
| 0.01 | 10.00 | 10.00 | 10.00 | 9.20 | 9.10 | 9.30 | 9.30 | 9.30 | 9.30 | 9.30 | 9.30 |
| 0.10 | 87.00 | 87.40 | 28.70 | 10.60 | 9.70 | 9.20 | 9.10 | 9.20 | 9.30 | 9.30 | 9.30 |
| 0.50 | 87.00 | 86.90 | 86.50 | 29.90 | 14.70 | 9.70 | 9.50 | 9.30 | 9.20 | 9.20 | 9.10 |
| 1.00 | 87.00 | 86.90 | 87.10 | 77.20 | 29.70 | 10.80 | 9.80 | 9.50 | 9.20 | 9.20 | 9.20 |
| 5.00 | 86.90 | 86.90 | 86.80 | 87.00 | 86.30 | 29.80 | 14.40 | 11.50 | 10.20 | 9.80 | 9.80 |
| 10.0 | 86.90 | 86.90 | 86.90 | 87.00 | 87.00 | 77.10 | 29.80 | 14.40 | 13.00 | 11.50 | 10.80 |
| 20.0 | 86.90 | 86.90 | 86.90 | 86.90 | 87.00 | 85.50 | 77.20 | 29.80 | 18.90 | 14.40 | 13.80 |
| 30.0 | 86.90 | 86.90 | 86.90 | 86.80 | 87.00 | 86.50 | 84.40 | **62.30** | 29.80 | 21.30 | 17.00 |
| 40.0 | 86.90 | 86.90 | 86.90 | 86.90 | 86.90 | 87.20 | 85.50 | 77.20 | 51.10 | 29.80 | 22.10 |
| 50.0 | 86.90 | 86.90 | 86.90 | 86.90 | 86.80 | 87.00 | 86.30 | 82.80 | 70.90 | 45.10 | 29.80 |

