# OpenReview forum: "Rethinking The Dependence Between Gradients and The Initial Point in Deep Learning"
_ICLR.cc/2024/Conference — ICLR 2024 Conference Withdrawn Submission_

### Official Review · Reviewer_izxw · 2023-10-26

**Soundness:** 1 poor
**Presentation:** 1 poor
**Contribution:** 1 poor
**Rating:** 1
**Confidence:** 4

**Summary:**

This work studies the transferability of cumulative gradients from one initialisation to another initialisation.
The transfer is done by upscaling the cumulative gradients and adding them to the other initialisation.
Experiments show that this transfer results in surprisingly good training accuracies.
Furthermore, the authors propose to additionally add the original initialisation in the transfer to obtain model fusion.
Experiments show that model fusion improves accuracies even more.
Finally, model fusion is applied to counter catastrophic forgetting.

**Strengths:**

I think I was able to understand the paper.

**Weaknesses:**

- (originality) This manuscript fails to mention literature on random feedback alignment (Lilicrap et al., 2016), where random gradients are shown to enable learning.
 - (originality) The literature that addresses catastrophic forgetting has been completely ignored.
   Furthermore, I find it hard to believe that this is the first attempt to use weight aggregation to solve catastrophic forgetting.
   I do not have a reference at hand, but I would expect this to be a simple baseline.
 - (originality) Existing literature on catastrophic forgetting seems to be completely ignored.
 - (clarity) It is not clear how these results would help to open up the black box of deep learning.
 - (clarity) It is not clear how this work relates to some of the referenced literature (e.g. to universal function approximation, information bottlenecks, ...).
 - (clarity) I do not see what additional information each table should add to the already included figures.
 - (clarity) I do not understand how to read Table&nbsp;6. I also do not understand what the bold number should highlight.
 - (significance) The experiments on catastrophic forgetting are not compared to any existing methods.
 - (quality) The manuscript claims that adding cumulative gradients to a different initial point leads to useful networks.
   I would have expected this statement to imply that the transfer works with a scale of $\alpha = 1$.
   However, in most settings, $\alpha >> 1$ is required to get reasonable results.
   As a result, the new initialisation is likely overruled by the upscaled gradients and therefore merely acts as some noise.
   Therefore, I would suspect the resulting network to be similar to the original with upscaled weights and some additional noise.
   These networks should be similar to the original but probably suffer from the upscaling.
   The same argument holds for the proposed model fusion.

### Minor comments

 - Please, clean up the citations (for most citations, the authors should be inside of the parentheses)
 - There are quite some word repetitions (e.g. Equation equation, ...)

### References

 - Lillicrap, T. P., Cownden, D., Tweed, D. B., & Akerman, C. J. (2016).
   Random synaptic feedback weights support error backpropagation for deep learning.
   Nature Communications, 7(1), 13276. https://doi.org/10.1038/ncomms13276

**Questions:**

1. What happens if you transfer cumulative gradients to a zero-initialised network?

---

### Official Review · Reviewer_gRM6 · 2023-10-30

**Soundness:** 2 fair
**Presentation:** 2 fair
**Contribution:** 3 good
**Rating:** 3
**Confidence:** 4

**Summary:**

The paper studies the relationship between gradient and initial point in deep learning optimization. It reveals a counter-intuitive phenomenon - representations of cumulative gradients exhibit some independence from the initial point from which they are calculated. This means that the learned gradients can be assigned to other arbitrarily initialized networks while maintaining a similar representation to the original network. Model fusion can also transfer learned representations across models by assigning the weights of one network to another network. When learning sequential tasks, both gradient assignment and model fusion can mitigate catastrophic forgetting to a certain extent. Extensive experiments on convolutional networks and ResNet demonstrate that this phenomenon occurs. This challenges traditional non-convex optimization theory of neural networks and provides new insights into their "black box" nature. In summary, the main contribution of this article is to reveal an interesting counterintuitive phenomenon, provide theory and experiments to analyze it, demonstrate its potential to improve deep learning techniques and provide new insights into understanding and improving neural network optimization.

**Strengths:**

$\textbf{Originality}$: The paper presents a highly original and counter-intuitive phenomenon that the representation of cumulative gradients exhibits some independence from their initial computation point. This surprising finding defies traditional optimization theories and reveals deep insights into the inner workings of neural networks. The exploration of gradient assignment and model fusion as techniques to mitigate catastrophic forgetting is also novel.

$\textbf{Quality}$: This article uses multiple image classification data sets including CIFAR10, CIFAR100, Caltech256, etc., a simple convolutional neural network structure in the cumulative gradient distribution experiment, and a network with residual structure in the model fusion experiment.

$\textbf{Clarity}$: This article is written relatively smoothly and has a clear structure.

$\textbf{Significance}$: The implications of this work for further demystifying the "black box" nature of deep neural networks are potentially very significant. The results challenge existing assumptions and open up new research directions in neural network optimization and representation learning.

**Weaknesses:**

While this paper presents an intriguing phenomenon, a few aspects could be improved to enhance the completeness and clarity of the work.

Firstly, some additional details on the experimental methodology would further strengthen reproducibility. Specifically, providing the full specifications for initialization schemes, optimization techniques, and learning rates utilized would allow readers to better comprehend the empirical results. Furthermore, incorporating key theoretical derivations directly into the main text, rather than relegating them to the appendix, would improve readability. Overall, reorganizing portions of the paper could enhance the flow and explanations for the reader, like explaining the phenomenon before presenting formulations.

Secondly, expanding the experimental evaluations to encompass more diverse datasets and network architectures would better demonstrate the generalizability of the phenomenon. The current experiments focus heavily on CIFAR10 and CIFAR100 with simple convolutional and residual networks. Evaluating the effects across more tasks and models could further substantiate the claims.

Finally, deriving more rigorous theoretical analyses to elucidate the underlying reasons for the surprising phenomenon of gradient independence could elevate the contributions. Supplementing the empirical results with detailed proofs grounded in optimization theory would provide greater justification for the counter-intuitive findings.

In summary, this paper uncovers an intriguing phenomenon but could benefit from refinements to the presentation, expanded experimental validations, and strengthened theoretical explanations. Addressing these aspects would help realize the full potential and impact of this promising work. With revision, the study has the opportunity to make significant contributions to demystifying deep neural network optimization.

**Questions:**

$\textbf{Question 1}$: Can you compare the scale of the initial values of the parameters of the neural network and the scale of its training trajectory? If the initial value of the network parameters is small, but the training trajectory is very long, then it seems that the initial position is not so important.

$\textbf{Question 2}$: Can you explain why in Tables 2 and 4, the results when alpha is greater than 1 are better than when alpha is equal to 1? This also seems counter-intuitive.

$\textbf{Question 3}$: Can you elaborate a little bit on the details of your experiment?

---

### Official Review · Reviewer_vmiY · 2023-10-31

**Soundness:** 1 poor
**Presentation:** 1 poor
**Contribution:** 1 poor
**Rating:** 3
**Confidence:** 4

**Summary:**

The authors argue through experiments that the cumulative gradients in neural networks have some degree of invariance from the initial point. They exploit this observation to train models on different initial points, and also to transfer weights among models (with the same architecture) trained on different datasets. The authors propose that their method can be used to mitigate catastrophic forgetting.


I tend towards rejection, based on the following main points.
- Presentation. The presentation of what is done is not clear. The main definitions and descriptions are so vague, that even the precise content of the main contribution must be guessed. I elaborate more on this in the weaknesses section.
- Soundness. Since the procedures are not described in detail and the code is not provided, it is not possible to assess the soundess of the contribution.
- Contribution. Although the authors state that they performed extensive simulations, I would say that they are instead pretty limited. For example, the main result, i.e. the independence of the initial conditions, is only shown on a single simple convolutional network architecture. This is not enough for a paper which contains no theory predicting the phenomenon (e.g. I would expect runs both on MLPs, on convnets with varying architectural details, and on large-scale models). Also, the method depends on a hyperparameter alpha, which adds to the usual hyperparameters and needs to be found by grid search.

**Strengths:**

If confirmed, the independence of the gradients from the initial point would challenge lots of established theory on the initialization of deep neural networks.

**Weaknesses:**

Presentation.
- The first page is boilerplate text which could apply to any paper, and does not add anything.
- The rest of the introduction feels vague and not to the point. There are several references here and there, but they do not help the readability of the paper, and it is not easy to follow the narrative.
- There is no mention to catastrophic forgetting in the intro (beyond the bullet point where the authors state they study it) nor in the related work section.
- The argued invariance is about cumulative gradients, but cumulative gradients are never explicitly defined, so essentially we don't know what we're talking about.
One guess would be that this is about gradient accumulation, but in this case all the accumulated gradients are used as a single gradient (i.e. it's just a way to circumvent memory problems). Eqs.4 and 5 hint at a definition, but they use a compact notation which is not self-explanatory and is not explained in detail by defining the objects or making the dependencies explicit. The way the equations are written, it would seem that the cumulative gradients are just the full-batch gradients calculated at step 1, but then the authors would just have said that, or have written that the theory is for GD instead of SGD (and also, it would not be clear what is done at the following steps). If instead we are not talking of full-batch gradients, it could be that we are talking of several gradients calculated at different times, but then there should be a dependence on time on the loss function and on the gradients. So essentially I am not sure about what the found symmetry consists of, and I do not want to be guessing.
My first suggestion would be to explicitly include time in the equation, and try to make it very clear what the preocedure consists of.

 - The procedure of assigning the waits from a differently initialized model is not clear. For example, are the models trained in parallel and the gradients assigned to the other model? Also, I would have appreciated attempts at really understanding the process. For example, what happens if I permutate the labels in the original model?

 - It is also not clear what is done when the authors say that weights from a model trained on another dataset are taken. What does it mean that the authors took a "well-trained" model from another dataset? How are the gradients transferred? At every time step? Is the procedure invariant under label permutation and weight permutation? Since the other datasets are taken, that have a different number of classes, also the numer of weights are different, so how are the weights in the last layer transferred.

 - It is not clear how models were trained, whether and how hyperparameter tuning was done, for how many epochs the models were trained and whether this was enough for convergence, what the loss functions and the optimizers were, and so on. These are important questions, both because of reproducibility, and because these elements could have an impact on whether the claimed phenomena are present.


- It is not clearly presented that the performances are training performances, and other times it is ultimately not clear. For example, table 1 mentions that the accuracy of the original model ($\theta_A$) is accuracy=1 (so I would guess we're talking about training accuracy), while table 2 says that the test accuracy of the original model is 0.61.
- The language looks sometimes inadequate. For example, "we employed the original convolutional network as our network architecture", or talking of "The conclusions in Equation equation 4 have been supported by numerous empirical results Nguyen & Hein (2017); Li et al. (2018)". Eq.4 does not have any conclusions, and it is not clear what the two references are cited for.

- the figures would be clearer if it was always explicit on the label whether training or test loss is depicted, and if the xlabels (alpha) were explicitly written.

- It should be specified on the captions what the numbers in tables 5 and 6 are (accuracies). Also, I suggest that all accuracies be written with the same notation (in previous tables they are written as numbers between 0 and 1, and now between 0 and 100).


Code.
- The code is not made available. This is especially a problem because I was not able to check and understand what the procedures consisted of.

**Questions:**

- I do not understand the reason of studying the small and medium version of the datasets, instead of just focusing on the whole dataset. I think it distracts, and is potentially misleading, since the more or less high training performances

- Why is the test accuracy in Fig.3 higher than the training accuracy, even though no data augmentation was used?

- In Eq.7, wouldn't it be more intuitive to write it as a linear interpolation between the two models [$\theta_n(1-\alpha)+\alpha\theta_A$]?

- From section 3 (and from intuition) we know that if we use the original model ($\theta_A$) we have better performance than if we transfer the gradients. So isn't it obvious that if we combine the transferred weights with the original ones we will have a better performance?

---

### Official Review · Reviewer_gVxp · 2023-10-31

**Soundness:** 2 fair
**Presentation:** 2 fair
**Contribution:** 1 poor
**Rating:** 3
**Confidence:** 5

**Summary:**

Cumulative gradients $g_A$ or solutions $\theta_A$ from training a neural network on task $A$ are scaled (by $\alpha$) and linearly combined with either randomly initialized parameters $\theta_i$ or the network $\theta_B$ from training on another task $B$. At sufficiently large $\alpha$ the resulting network is able to recover a large part of the performance of the original network $A$. This result is applied to mitigate catastrophic forgetting by a linear combination of networks independently trained on tasks $A$ and $B$.

**Strengths:**

This work proposes an interesting idea to decouple initialization from training, which challenges some intuitions built by previous works (e.g. lottery ticket hypothesis). The application to catastrophic forgetting is supported by a theoretical derivation which points out potential directions for improving such algorithms. The paper is well structured and easy to follow.

**Weaknesses:**

The main issue is that this work can be recast as adding isotropic random noise to a trained neural network, which is a common procedure in literature such as neural network generalization or adversarial attack.
- Models at initialization ($\theta_i$) are standard Gaussians and $\theta_A = \theta_1 + g_A$, so $\theta_0 + \alpha g_A$ or $\theta_0 + \alpha \theta_A$ are equivalent to a noisy perturbation of a trained network.
- Since ReLU networks are invariant to pre-activation scaling, the only effect of $\alpha$ on the network is to change the relative scale of the biases in the linear and batchnorm layers. As this work only investigates shallow networks there may not be enough layers for this to worsen the model's outputs at values of $\alpha$ near 1. Thus, scaling $\alpha$ probably gives similar results to scaling the added perturbation noise $\theta_i$.
- There is a lack of comparison with other methods from the model fusion and catastrophic learning literature. For example, [
Model Fusion via Optimal Transport (Singh \& Jaggi, 2020)](https://proceedings.neurips.cc/paper/2020/hash/fb2697869f56484404c8ceee2985b01d-Abstract.html) fuses larger models while preserving more accuracy when moving between disjoint datasets.

The experiments could be fleshed out more (see questions), and some of the presentation could also be improved:
- the introduction and related work section are overly broad and the paper does not make clear how some of the cited works are related, such as the work related to information compression (Tishby et al.).
- the interesting results in appendix B could be shown in the main text instead.

**Questions:**

1. What are the baseline performances of the trained networks, such as $\theta_A$ without adding gradients/fusion, as well as performance of at different scales, i.e. $\alpha \theta_A$?
2. What are the results for both residual and non-residual networks in all of the experiments? Some larger networks (e.g. VGG-16, ResNet-18, etc.) could also be shown.
3. Does the linear combination of $\alpha \theta_A + \beta \theta_B$ preserve performance on task $B$ as well as task $A$? In general, if combining two networks which are both trained, it would be instructive to know how the performance on both tasks varies with $\alpha$.
4. Given that this work investigates the performance of networks along linear paths, how does this work relate to the literature investigating the linear (mode) connectivity of networks?
5. What are some other implications or applications of this work?